# Viral load and contact heterogeneity predict SARS-CoV-2 transmission and super-spreading events

**Ashish Goyal[1], Daniel B Reeves[1], E Fabian Cardozo-Ojeda[1], Joshua T Schiffer[1,2,3]\*, Bryan T Mayer[1]**

[1]Vaccine and Infectious Diseases Division, Fred Hutchinson Cancer Research Center, Seattle, United States; [2]Department of Medicine, University of Washington, Seattle, United States; [3]Clinical Research Division, Fred Hutchinson Cancer Research Center, Seattle, United States

**Abstract** SARS-CoV-2 is difficult to contain because many transmissions occur during pre-symptomatic infection. Unlike influenza, most SARS-CoV-2-infected people do not transmit while a small percentage infect large numbers of people. We designed mathematical models which link observed viral loads with epidemiologic features of each virus, including distribution of transmissions attributed to each infected person and duration between symptom onset in the transmitter and secondarily infected person. We identify that people infected with SARS-CoV-2 or influenza can be highly contagious for less than 1 day, congruent with peak viral load. SARS-CoV-2 super-spreader events occur when an infected person is shedding at a very high viral load and has a high number of exposed contacts. The higher predisposition of SARS-CoV-2 toward super-spreading events cannot be attributed to additional weeks of shedding relative to influenza. Rather, a person infected with SARS-CoV-2 exposes more people within equivalent physical contact networks, likely due to aerosolization.

\*For correspondence:
jschiffe@fredhutch.org

## Introduction

The SARS-CoV-2 pandemic is an ongoing tragedy that has caused nearly 2 million deaths and massively disrupted the global economy. The pandemic is rapidly expanding in the United States and is re-emerging focally in many countries that had previous success in limiting its spread (https://corona-virus.jhu.edu/map.html).

Two features have proven challenging in containing outbreaks. First, most transmissions occur during the pre-symptomatic phase of infection (*He et al., 2020*; *Moghadas et al., 2020*; *Tindale et al., 2020*). Underlying this observation is a highly variable incubation period, defined as time between infection and symptom onset, which often extends beyond an infected person's peak viral shedding (*Ganyani et al., 2020*).

Second, there is substantial over-dispersion of the secondary infection distribution (individual $R_0$) for an individual infected with SARS-CoV-2 (*Endo et al., 2020*). An over-dispersed $R_0$ means that most infected people do not transmit at all (individual $R_0 = 0$) while a minority of infected people are super-spreaders (individual $R_0 > 5$). If the average population $R_0$ is greater than 1, then exponential growth of cases occurs in the absence of effective interventions (*Lloyd-Smith et al., 2005*). Overdispersion has been quantified: approximately 10–20% of infected people account for 80% of SARS-CoV-2 transmissions (*Endo et al., 2020*; *Bi et al., 2020*). SARS-CoV-2 super-spreader events, in which the duration of contact between a single transmitter and large number of secondarily infected people is often limited to hours, are well documented (*Hamner et al., 2020*; *Park et al., 2020*).

This pattern is not evident for influenza which has more homogeneous individual transmission (*Cowling et al., 2009*; *Brugger and Althaus, 2020*). Differing viral load kinetics between the two viruses might explain this distinction; SARS-CoV-2 is often present intermittently in the upper airways for many weeks (*Qi et al., 2020*; *Cao et al., 2020*), while influenza is rarely shed for more than a week (*Pawelek et al., 2012*). Alternatively, SARS-CoV-2 aerosolization may effectively increase the number of people with true viral exposures given the same contact network. This means that a SARS-CoV-2-infected person in a crowded indoor space could lead to more transmissions relative to an influenza-infected person.

Viral load is recognized as a strong determinant of transmission risk (*Watanabe et al., 2010*). For influenza, the dose of viral exposure is related to the probability of infection in human challenge studies (*Memoli et al., 2015*) and early antiviral treatment reduces household transmission (*Pebody et al., 2011*; *Goldstein et al., 2010*). Household shedding of human herpesvirus-6 is closely linked to subsequent infection in newborns (*Mayer et al., 2020*) and infants shedding high levels of cytomegalovirus in the oropharynx predictably transmit the virus back to their mothers (*Boucoiran et al., 2018*). Studies in mice definitively demonstrated that viral exposure dose determines likelihood of SARS-CoV-1 infection, (*Watanabe et al., 2010*) and SARS-CoV-2 experiments in golden hamsters are also highly suggestive of dose-dependent infection (*Sia et al., 2020*).

The epidemiology of viral infections can also be perturbed by biomedical interventions that lower viral load at mucosal transmission surfaces. Reduction of genital herpes simplex virus-2 shedding with antiviral treatments decreases probability of transmission (*Corey et al., 2004*). Suppressive antiretroviral therapy (ART) for HIV virtually eliminates the possibility of partner-to-partner sexual transmission and has limited community transmission dramatically (*Rodger et al., 2019*; *Cohen et al., 2016*).

SARS-CoV-2 overdispersion, aerosolization, and dose-dependent infection all require urgent attention as possible avenues to understand and mitigate the pandemic as it continues to wreak havoc. Early therapies that lower peak viral load may reduce the severity of COVID-19 but may also decrease the probability of transmission and super-spreader events (*Schiffer et al., 2020*). Similarly, the effectiveness of policies such as limiting mass gatherings, and enforcing mask use can be directly evaluated by their ability to reduce exposure viral load and transmission risk (*Leung et al., 2020*). Here, we developed a transmission simulation framework to capture the contribution of viral load to observed epidemiologic transmission metrics for influenza and SARS-CoV-2 and used this approach to explain why SARS-CoV-2 is predisposed to super-spreading events.

## Results

### Overall approach

We designed a series of steps to estimate the viral load required for SARS-CoV-2 and influenza transmission, as well as conditions required to explain the observed over- dispersion of secondary infections (*individual $R_0$*) and frequent super-spreader events associated with SARS-CoV-2 but not influenza. This process included within-host modeling of viral loads, simulations of exposures and possible transmissions based on various transmission dose response curves, testing of various parameter sets against epidemiologic data and exploratory analyses with the best fitting model (*Figure 1*, *Figure 1—figure supplement 1*).

### Within-host mathematical model of SARS CoV-2 shedding

First, we used our previously developed within-host mathematical model (equations in the Materials and methods), (*Goyal et al., 2020*) to generate plausible viral load patterns in the upper airway of an infected person or *transmitter* who could potentially transmit the virus to others (*Figure 1*, *Figure 1—figure supplement 2a*). Briefly, the model captures observed upper airway viral kinetics from 25 people from four different countries (*Wölfel et al., 2020*; *Lescure et al., 2020*; *Young et al., 2020*; *Kim et al., 2020*). Key observed features include an early viral peak followed by a decelerating viral clearance phase, which in turn leads to a temporary plateau at a lower viral load, ultimately followed by rapid viral elimination. Our model captures these patterns by including a density-dependent term for early infected cell elimination and a nonspecific acquired immune term for late infected cell elimination.

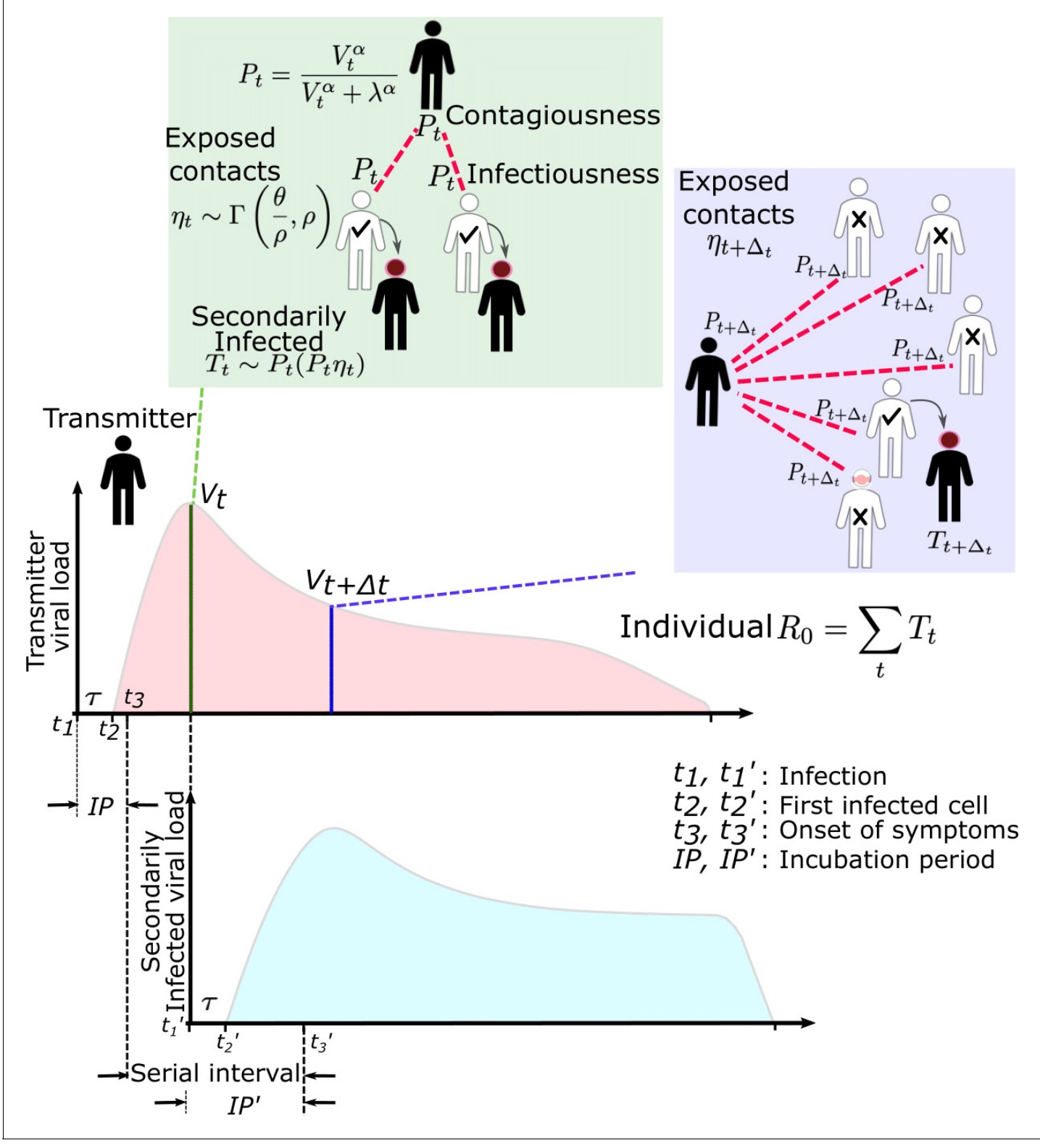

**Figure 1.** SARS-CoV-2 and influenza transmission model schematic. In the above cartoon, the transmitter has two exposure events at discrete timepoints resulting in seven total exposure contacts and three secondary infections. Transmission is more likely at the first exposure event due to higher exposure viral load. To model this process, the timing of exposure events and number of exposed contacts is governed by a random draw from a gamma distribution which allows for heterogeneity in number of exposed contacts per day (*Figure 1—figure supplement 3*). Viral load is sampled at the precise time of each exposure event. Probability of transmission is identified based on the product of two dose curves (*Figure 1—figure*

*Figure 1 continued on next page*

*Figure 1 continued*

*supplement 2c,d*) which capture contagiousness (probability of viral passage to an exposure contact's airway) and infectiousness (probability of transmission given viral presence in the airway). Incubation period (*Figure 1—figure supplement 4*) of the transmitter and secondarily infected person is an input into each simulation and is depicted graphically. Individual $R_0$ is an output of each simulation and is defined as the number of secondary infections generated by an infected individual. Serial interval is an output of each simulated transmission and is depicted graphically.

The online version of this article includes the following figure supplement(s) for figure 1:

**Figure supplement 1.** Mathematical model workflow.

**Figure supplement 2.** Mathematical model of SARS-CoV-2 transmission dynamics.

**Figure supplement 3.** Stochastic simulations of the number of exposure contacts over time for varying dispersion (ρ).

**Figure supplement 4.** Gamma distribution functions of incubation periods.

One limitation of our model is that only half of study participants provided longitudinal viral load data from the very early days of infection when COVID-19 is often pre-symptomatic. Therefore, the model's output is most reliable for later time points. In particular, we have somewhat limited information on viral expansion rate and duration of peak shedding. To impute possible variability, we generated a set of heterogeneous shedding curves in which the viral upslope, the downslope of viral load after peak and the viral load during plateau phase were varied (*Figure 1—figure supplement 2b*). Overall, the model generated several distinct patterns of infection: rapid elimination after the initial peak, a prolonged plateau phase with a low viral load, and a prolonged plateau phase with higher viral load. We simulated the transmission model with and without imputed heterogeneity.

## Transmission dose response curves

We defined an *exposure event* in very specific biologic terms as a discrete event consisting of sufficient contact in time and space between a transmitter and one or more uninfected persons (*exposure contacts*) to allow for the possibility of a successful transmission. An *exposure contact* is defined as a susceptible person who is exposed to a SARS-CoV-2-infected person for a sufficient period of time and at a close enough distance to allow for the possibility of a successful transmission, which is then determined by the viral load of the infected person. The number of exposure contacts can in theory be reduced by social distancing measures, quarantine, or masking.

We next designed hundreds of dose response curves which separately predict contagiousness (CD curves) and infectiousness (ID curves) at a certain viral dose given an exposure contact. *Contagiousness* is defined as the viral-load-dependent probability of passage of virus-laden droplets or airborne particles from the airways of a potential transmitter to the airway of an exposure contact. *Infectiousness* is defined as the viral-load-dependent probability of transmission given direct airway exposure to virus in an exposure contact. *Transmission risk* is the product of these two mechanistic probabilities derived from the ID and CD curves and results is a transmission dose (TD) response curve. Each CD or ID curve is defined by its ID50 (λ) or viral load at which contagion or infection probability is 50% (*Figure 1—figure supplement 2c*), as well as its slope (α) (*Figure 1—figure supplement 2d*; *Brouwer et al., 2017*). The TD50 is defined as viral load at which there is 50% transmission probability. We assumed equivalent curves for contagiousness and infectiousness for model fitting purposes. We also considered a simpler model with only a single TD curve (for *infectiousness*) and obtained qualitatively similar results (Materials and methods). Of note, a null model in which there is an assumed fixed probability of infection at all timepoints during infection poorly fit the observed data.

## Exposure contact rate simulations

We introduced heterogeneity of exposure contact rates among possible transmitters by randomly selecting from a gamma distribution defined by mean number of exposure contacts per day (θ) and a scaling factor (*ρ*) that controls daily variability (*Figure 1—figure supplement 3*).

## Transmission simulations

For each defined exposure contact, viral load in the transmitter was sampled and transmission risk was then identified based on the product of the CD and ID curves, or the TD curve (*Figure 1—figure supplement 2e,f*; *Figure 1*). Based on these probabilities, we stochastically modeled whether a

transmission occurred for each exposure contact. This process was repeated when there were multiple possible exposure events within a given discretized time interval and the total number of exposures and transmissions within that interval was calculated.

For each successful transmission, we assumed that it takes $\tau$ days for the first infected cell to produce virus. To inform simulated values of *serial interval* (SI or time between symptom onset in the secondarily infected and transmitter), we randomly selected the *incubation period* (IP), for both the transmitter and the newly infected person, from a gamma distribution based on existing data (*Figure 1—figure supplement 4a*). (*Ganyani et al., 2020*; *Lauer et al., 2020*) Incubation period was defined as time from infection to the time of the onset of symptoms, where the mean incubation for SARS-CoV-2 is 5.2 days compared to 2 days for influenza (*Ganyani et al., 2020*; *Cowling et al., 2009*; *Lauer et al., 2020*).

## Model fitting

In order to identify the parameter set that best recapitulated the observed data, we first performed a grid search simulating 41,7792 parameter sets with 256 possible TD curves defined by ID50 and CD50 ($\lambda$) and slope ($\alpha$), along with 408 combinations of the mean exposed contact rate per day ($\theta$) and associated variance parameter ($\rho$), and values of $\tau \in [0.5, 1, 2, 3]$ days. We aimed to identify the parameter set that best recapitulated the following features of the observed epidemiologic and individual-level data for SARS-CoV-2: mean $R_0$ across individuals ($R_0 \in [1.4, 2.5]$), (*Ganyani et al., 2020*; *Endo et al., 2020*; *Bi et al., 2020*; *Du et al., 2020*; *World Health Organization, 2020*) mean serial interval across individuals (SI $\in [4.0, 4.5]$), (*Ganyani et al., 2020*; *Du et al., 2020*; *Nishiura et al., 2020*) cumulative distribution functions of individual $R_0$, (*Endo et al., 2020*; *Bi et al., 2020*; *Zhang et al., 2020*; *Dillon, 2020*; *Miller, 2020*) and cumulative distribution functions of serial intervals derived from SARS-CoV-2 transmission pair studies that were conducted early during the pandemic (*Du et al., 2020*), prior to any confounding influence of social distancing measures with the exception of likely post-symptomatic self-isolation behavior. Here, we define *individual $R_0$* as the total number of secondary transmissions from the transmitter in a fully susceptible population (Materials and methods). Given that viral RNA is composed mostly of non-infectious material, we further checked the closeness of the solved ID curve with the observed relationship between viral RNA and probability of positive viral culture from a longitudinal cohort of infected people (*van Kampen, 2020*).

## Influenza modeling

Next, we performed equivalent analyses for influenza (results detailed below and in Figures 6–8) to explain the lower frequency of observed super-spreader events with this infection. Influenza viral kinetics were modeled using a previously data-validated model (*Baccam et al., 2006*). Incubation periods for influenza are lower and less variable than for SARS-CoV-2 and were randomly selected for each simulation of the model using a gamma distribution (*Figure 1—figure supplement 4b*; *Lessler et al., 2009*). We again fit the model to: mean $R_0$ across individuals ($R_0 \in [1.1, 1.5]$), (*Opatowski et al., 2011*; *Cowling et al., 2010*; *Roberts and Nishiura, 2011*) mean serial interval (SI $\in [2.9, 4.3]$), (*Cowling et al., 2009*) cumulative distribution functions of individual $R_0$ corresponding to the 2008–2009 influenza A H1N1 epidemic season with mean $R_0 = 1.26$ and dispersion parameter = 2.36 in the negative binomial distribution, and cumulative distribution functions of serial intervals (*Cowling et al., 2009*; *Brugger and Althaus, 2020*; *Opatowski et al., 2011*).

## Model-predicted individual $R_0$ and serial intervals for SARS-CoV-2 infection

A single model parameter set ([$\alpha, \lambda, \tau, \theta, \rho$] = [0.8, $10^7$, 0.5, 4, 40]) most closely reproduced empirically observed individual $R_0$ and serial interval histograms (*Figure 2a,c*) and cumulative distribution functions (*Figure 2b,d*).

The preciseness of the estimated parameter set was also independently confirmed with the use of Approximate Bayesian Computation (ABC) rejection sampling method, (*Liepe et al., 2014*) and a finer grid search in proximity to the aforementioned optimal solution (shown in Materials and methods). Despite assuming that each infected person sheds at a high viral load for a period of time (*Figure 1*, *Figure 1—figure supplement 2b*), the model captured the fact that ~75% of 10,000

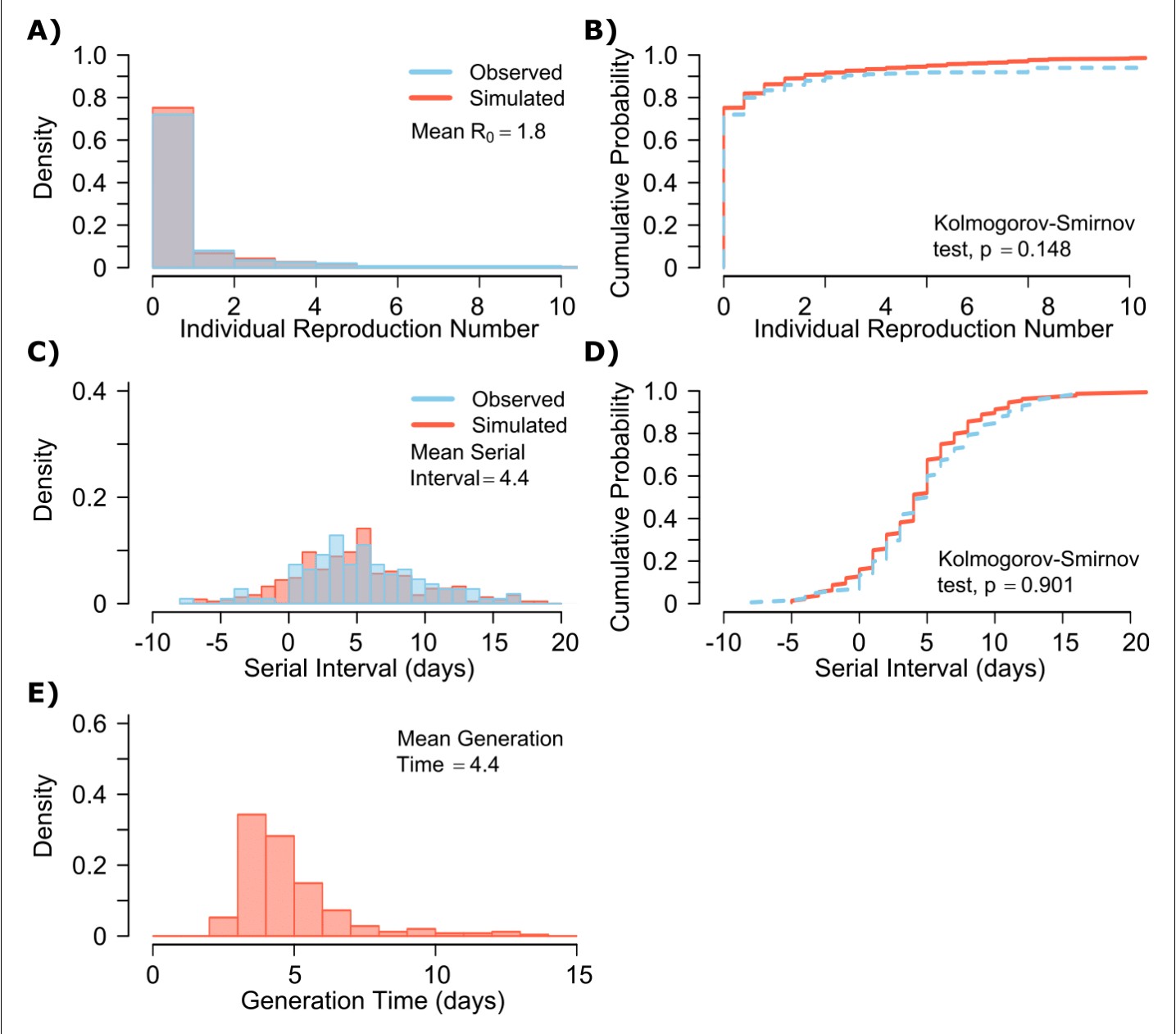

**Figure 2.** SARS-CoV-2 transmission model fit. (A) Simulated and actual frequency histograms of individual $R_0$ values, (***Endo et al., 2020***) (B) Simulated and actual cumulative distribution of individual $R_0$ values. (C) Simulated and actual frequency histograms of individual serial intervals, (***Du et al., 2020***) (D) Simulated and actual cumulative distribution of individual serial intervals. (E) Frequency distribution of simulated generation times.
The online version of this article includes the following figure supplement(s) for figure 2:

**Figure supplement 1.** Mathematical model recapitulation of relationship between SARS-CoV-2 viral load and viral culture.

simulated transmitters do not infect any other people and that each increase in the number of individual transmissions is associated with a lower probability (*Figure 2a*). Furthermore, we also fit the model to a higher population $R_0$ of 2.8-2.9 and arrived at a similar set of parameter values but with a higher daily rate of exposure contacts ([$\alpha$, $\lambda$, $\tau$, $\theta$, $\rho$] = [0.8, $10^{7.5}$, 0.5, 20, 30]) confirming the robustness of the qualitative results of the model.

SARS-CoV-2 viral load was recently measured with viral RNA levels and mapped to concurrent probability of positive viral culture in a Dutch cohort (***van Kampen, 2020***). Our model output demonstrated a nearly equivalent infectious dose response curve if we multiplied modeled viral RNA

levels by 25 (*Figure 2—figure supplement 1*): this adjustment was likely necessary because viral loads in the Dutch study participants were notably higher than those in German, Singaporean, Korean, and French participants used in our intra-host model fitting, perhaps due to different sampling technique or primers used in PCR assays (*Wölfel et al., 2020*; *Lescure et al., 2020*; *Young et al., 2020*; *Kim et al., 2020*; *van Kampen, 2020*).

The model also generated super-spreader events with 10,000 simulated transmissions (*Figure 2b*). If super-spreaders are defined as those who produce at least five secondary infections, we estimate that ~10% of all infected people and ~35% of all transmitters are super-spreaders. If super-spreaders are defined as those who produce at least 10 secondary infections, we estimate that ~6% of all infected people and ~25% of all transmitters are super-spreaders. If super-spreaders are defined as those who produce at least 20 secondary infections, we estimate that ~2.5% of all infected people and ~10% of all transmitters are super-spreaders. If super-spreaders are defined as those producing $\geq 5$, $\geq 10$, or $\geq 20$ secondary infections, the contribution to all secondary infections is estimated at ~85%, ~70%, or ~44%, respectively (*Table 1*).

The model also recapitulated the high variance of the serial interval observed within SARS-CoV-2 transmission pairs, including negative values observed in the data (*Figure 2c,d*). We next projected *generation time*, defined as the period between when an individual becomes infected and when they transmit the virus, for all transmission pairs and identified that the mean serial interval (4.4 days) provides an accurate approximation of mean generation time. However, the variance of generation time was considerably lower and by definition does not include negative values. A majority of generation times fell between 4 and 7 days, compared to −5 to 12 days for the serial interval (*Figure 2e*).

## Viral load thresholds for SARS-CoV-2 transmission

The optimized ID curve has an ID50 of $10^7$ viral RNA copies and a moderately steep slope (*Figure 3a*). The TD50 for SARS-CoV-2 was slightly higher at $10^{7.5}$ viral RNA copies (*Figure 3a*). To assess the impact of these parameters on transmission, we performed simulations with 10,000 transmitters and concluded that transmission is very unlikely (~0.00005%) given an exposure to an infected person with an upper airway viral load of $<10^4$ SARS-CoV-2 RNA copies, and unlikely (~0.002%) given an exposure to an infected person with a viral load of $<10^5$ SARS-CoV-2 RNA copies. On the other hand, transmission is much more likely (39%) given an exposure to an infected person who is shedding $>10^7$ SARS-CoV-2 RNA copies, and 75% given an exposure to an infected person with a viral load of $>10^8$ SARS-CoV-2 RNA copies. We obtain similar results (not shown) when we solve our model using the assumption of homogeneous viral load trajectories (according to viral take off time, peak viral load and first phase decline) as in *Figure 1—figure supplement 2a*.

## Narrow duration of high infectivity during SARS-CoV-2 infection

We next plotted the probability of infection given an exposure to a transmitter. Under multiple shedding scenarios, the window of high probability transmission is limited to time points around peak viral load, and some heterogeneity in regard to peak infectivity is noted between people (*Figure 3b–d*). In general, infected persons are likely to be most infectious (i.e., above TD50) for a ~0.5–1.0 day period between days 2 and 6 after infection. We therefore conclude that the observed wide variance in serial interval (*Figure 2c*) results primarily from the possibility of highly discrepant

**Table 1.** Prevalence of super-spreaders among transmitters, and contribution of super-spreading events to all SARS-CoV-2 and influenza transmissions.

Estimates are from 10,000 simulations.

| Super-spreader definitions | SARS-CoV-2 | | | Influenza | | |
|---|---|---|---|---|---|---|
| | All infected people | All transmitters | Contribution of super-spreaders to transmissions | All infected people | All transmitters | Contribution of super-spreaders to transmissions |
| Individual $R_0 \geq 5$ | ~10% | ~35% | ~85% | ~2% | ~3% | ~10% |
| Individual $R_0 \geq 10$ | ~6% | ~25% | ~70% | ~0% | ~0% | ~0% |
| Individual $R_0 \geq 20$ | ~2.5% | ~10% | ~44% | ~0% | ~0% | ~0% |

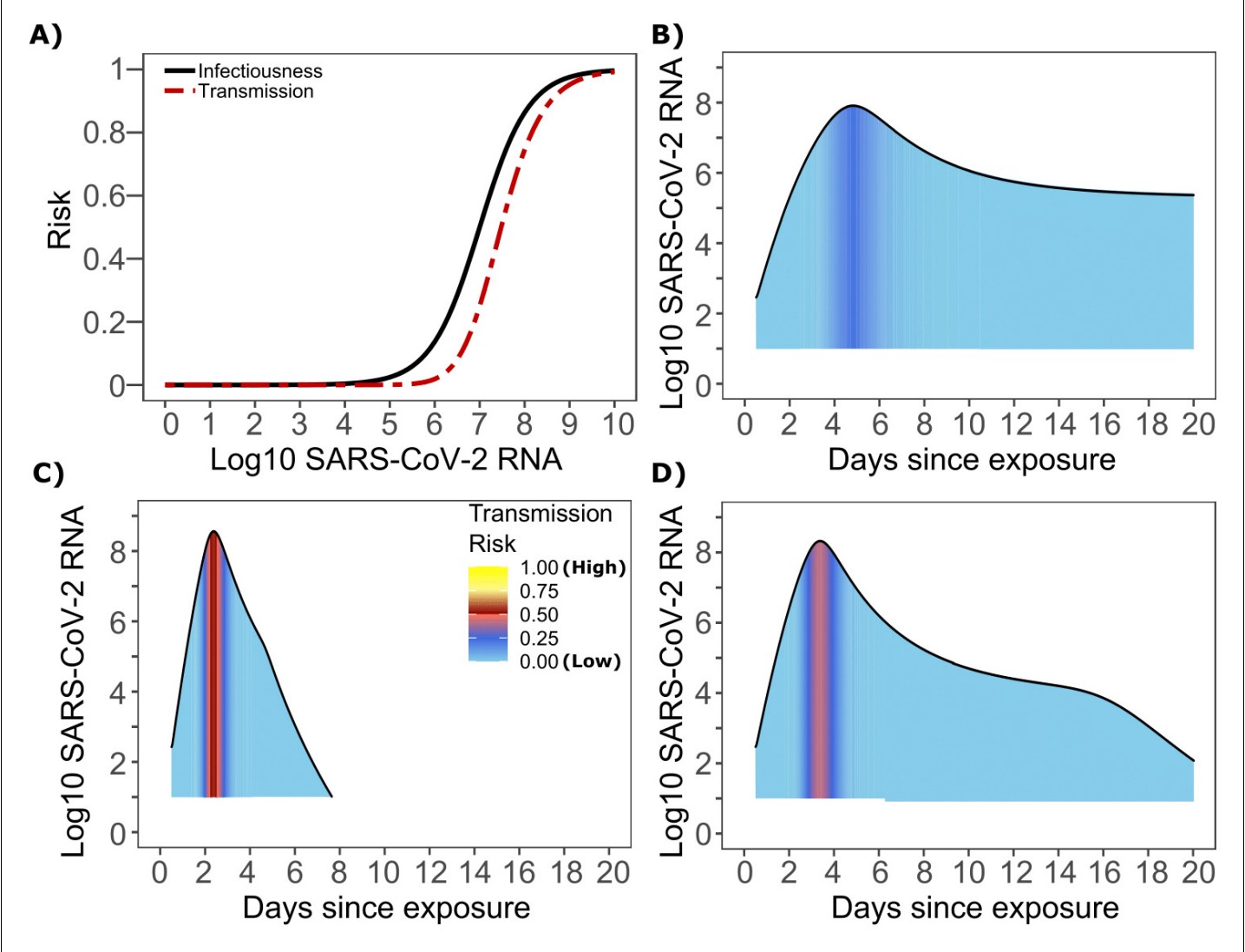

**Figure 3.** SARS-CoV-2 transmission probability as a function of shedding. (**A**) Optimal infectious dose (ID) response curve (infection risk = $P_t$) and transmission dose (TD) response curve (transmission risk = $P_t * P_t$) curves for SARS-CoV-2. Transmission probability is a product of two probabilities, contagiousness and infectiousness (**Figure 1**). (**B-D**) Three simulated viral shedding curves. Heat maps represent risk of transmission at each shedding timepoint given an exposed contact with an uninfected person at that time.

incubation periods between the transmitter and infected person, rather than wide variability in shedding patterns across transmitters.

## Requirements for SARS CoV-2 super-spreader events

The solved value for exposed contact network heterogeneity (ρ) is 40 indicating high variability in day-to-day exposure contact rates (**Figure 1—figure supplement 3d**) with an average number of exposed contacts per day (θ = 4). We generated a heat map from our TD curve to identify conditions required for super-spreader events which included viral load exceeding $10^7$ SARS CoV-2 RNA copies and a high number of exposure contacts on that day. We observed an inflection point between $10^6$ and $10^7$ SARS CoV-2 RNA copies, after which large increases in the number of daily exposure contacts prominently increases the number of transmissions from a single person (**Figure 4a**). The exposure contact network occasionally resulted in days with ≥150 exposure contacts per day, which may allow an extremely high number of secondary infections from a single person (**Figure 4a**).

We next plotted transmission events simulated on a daily basis over 30 days since infection, from 10,000 transmitters, according to viral load at exposure and number of exposure contacts on that

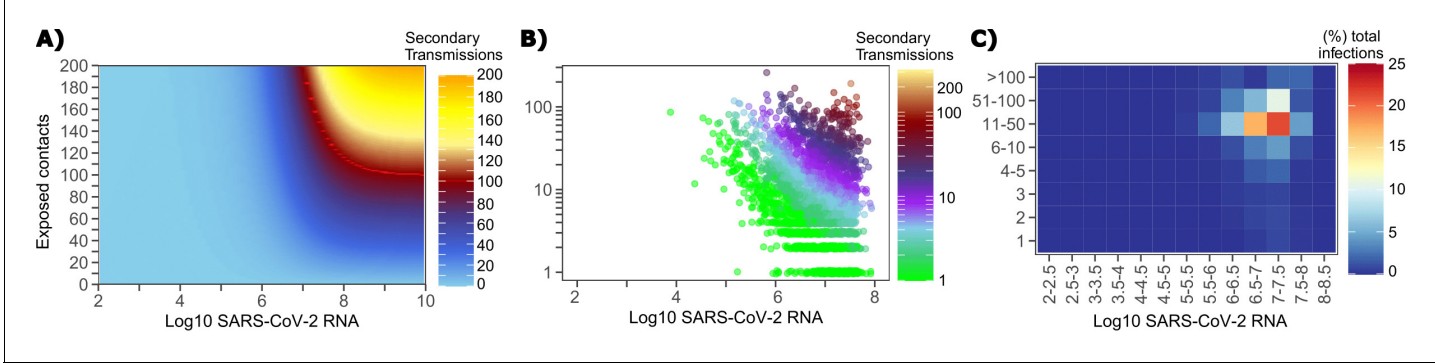

**Figure 4.** Conditional requirements for SARS-CoV-2 superspreading events. (**A**) Heatmap demonstrating the maximum number of feasible secondary infections per day from a transmitter given an exposure viral load on log10 scale (x-axis) and number of exposed contacts per day (y-axis). The exposed contact network governed by the gamma distribution allows for a range of values between 0–200 per day with values > 200/day outside of the 99.99% quantile. Such high exposure contacts per day are sufficient for multiple transmissions from a single person per day. (**B**) 10,000 simulated transmitters followed for 30 days. The white space is a parameter space with no transmissions. Each dot represents the number of secondary transmissions from a transmitter per day. Input variables are log10 SARS-CoV-2 on the start of that day and number of contact exposures per day for the transmitter. There were 1,154,001 total simulated exposure contacts and 15,992 total infections. (**C**) 10,000 simulated transmitters with percent of infections due to exposure viral load binned in intervals of 0.5 intervals on log10 scale (x-axis) and number of exposed contacts (y-axis).

day (*Figure 4b*). Secondary transmissions to only 1–3 people occurred almost exclusively with daily numbers of exposure contacts below 10 with any exposure viral load exceeding $10^6$ RNA copies or with higher numbers of exposure contacts per day and viral loads exceeding $10^5$ RNA copies. Massive super-spreader events with over 50 infected people almost always occurred at viral loads exceeding $10^7$ RNA copies with high levels of concurrent exposure contacts (*Figure 4b*).

We next identified that over 50% of simulated secondary infections were associated with a transmitter who has a high number of exposed contacts (11–100 per day) and a viral load exceeding $10^6$ RNA copies (*Figure 4c*), which is the mechanistic underpinning of why ~70% of all simulated secondary infections arose from transmitters who produced more than 10 secondary infections (*Table 1*).

## Longer duration of infectivity during the earliest phase of the pandemic in Wuhan

The serial interval was reported to be longer during the initial stages of the pandemic in Wuhan (before January 22, 2020, termed as pre-lockdown) possibly because infected people were initially less likely to self-isolate after developing symptoms (*Li et al., 2020*; *Ali et al., 2020*). Therefore, we refit our model to this data with mean serial interval of ~7.5 days, mean $R_0 \in [2.2, 2.5]$ and the individual $R_0$ distribution as shown in *Figure 1a*. The new parameter estimates ($[\alpha, \lambda, \tau, \theta, \rho] = [0.6, 10^{5.0}, 0.5, 0.7, 40]$) yields good fits to the distribution of individual $R_0$ as well as serial interval (*Figure 5a–b*). Importantly, the optimized ID curve has an ID50 of $10^5$ viral RNA copies during the early phase of the pandemic (pre-lockdown), ~100-fold lower than post-lockdown (*Figure 3a*). Keeping in mind that ID50 is an average measure, we propose that ID50 estimates leading to more prolonged and intimate exposure contacts that were of higher risk given an equivalent viral load, whereas estimates from later in the pandemic (*Figure 3*) reflect greater social distancing and higher use of masking. This is also evident in the longer duration of infectivity during pre-lockdown (*Figure 5d*). The predicted exposure contact network in Wuhan is highly dispersed and resembles *Figure 1—figure supplement 3d*.

## Model predicted individual $R_0$ and serial intervals for influenza infection

A single model parameter set most closely reproduced empirically observed histograms and cumulative distribution functions for individual $R_0$ and serial intervals for influenza: ($\alpha, \lambda, \tau, \theta, \rho$) = (0.7, $10^{5.5}$, 0-0.5, 4, 1). ID50 values for influenza were lower than SARS-CoV-2, but a direct comparison cannot be made because tissue culture infectious dose (TCID) has been more commonly used for measurements of influenza viral load, whereas viral RNA is used for SARS-CoV-2. Nevertheless, TCID

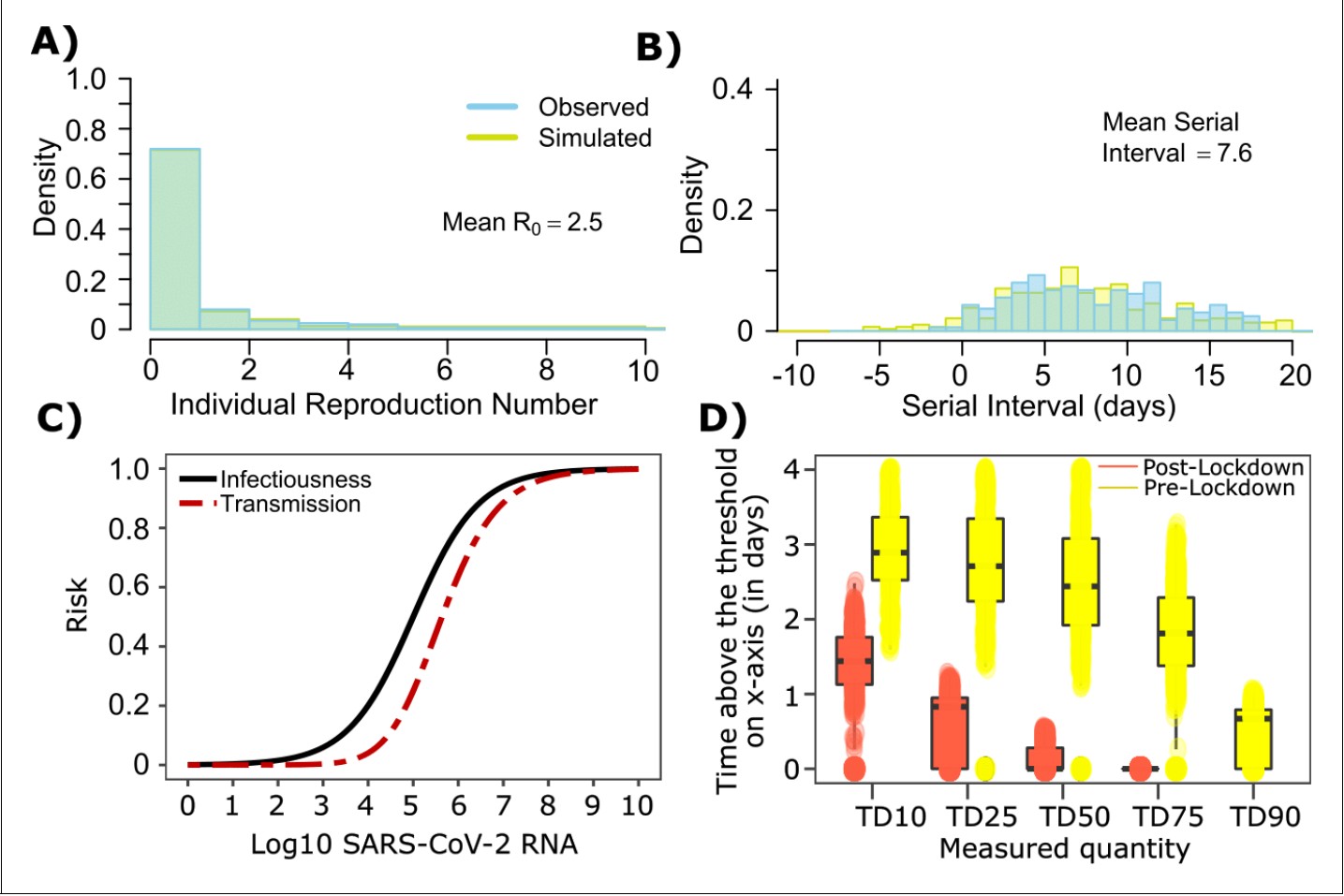

**Figure 5.** SARS-CoV-2 transmission model fit during the early phase of the pandemic in Wuhan. (A) Simulated and observed frequency histograms of individual $R_0$ values (*Endo et al., 2020*). (B) Simulated and actual frequency histograms of individual serial intervals (*Li et al., 2020*). (C) Optimal infectious dose (ID) response curve (infection risk = $P_t$) and transmission dose (TD) response curve (transmission risk = $P_t * P_t$) curves for SARS-CoV-2. (D) Boxplots of duration of time spent above TD10, TD25, TD50, TD75 and TD90 for 10,000 simulated SARS-CoV-2 infection during the early phase of pandemic (termed as pre-lockdown, before January 22, 2020) and post-lockdown (after January 22, 2020). TD10, TD25, TD50, TD75, and TD90 are viral loads at which transmission probability is 10%, 25%, 50%, 75%, and 90% respectively. The midlines are median values, boxes are interquartile ranges (IQR), and datapoints are outliers.

is a closer measure of infectious virus and it is thus reasonable that ID50 based on TCID for influenza would be ~30-fold lower than ID50 based on total viral RNA (infectious and non-infectious virus) for SARS-CoV-2 (*van Kampen, 2020*).

The other notable difference was a considerably lower $\rho$ value for influenza (*Figure 1—figure supplement 3b*), denoting much less heterogeneity in the number of exposure contacts per person while the average daily exposure contact was the same for both viruses (4 per day). The model captures the fact that 40% of people in this influenza infected cohort do not transmit to anyone else and that each increase in the number of individual transmissions is associated with a lower probability (*Figure 6a*). In keeping with the observed data, our model simulations predicted that relative to SARS-CoV-2, super-spreader events involving five or more people were ~5-fold less common overall and 10-fold less common among transmitters (~2% of all infected people and ~3% of transmitters) (*Figure 6b*, *Table 1*). Super-spreaders defined as those infecting ≥5 individuals contributed to only ~10% to all transmissions (*Table 1*).

The model also recapitulated the lower variance of serial interval for influenza relative to SARS-CoV-2 (*Figure 6c,d*). We next identified that the mean and variance of the serial interval provide

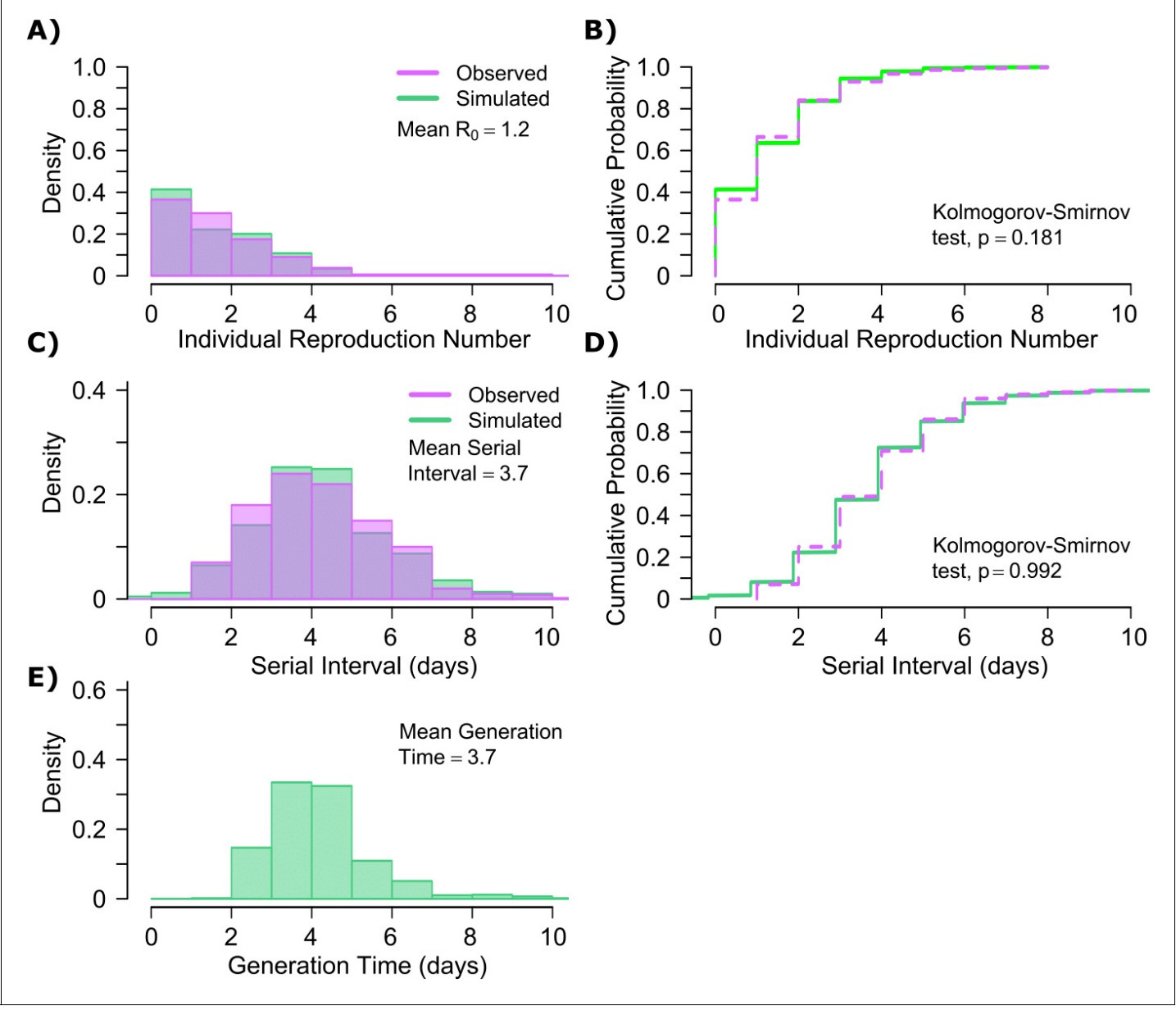

**Figure 6.** Influenza transmission model fit. (A) Simulated and actual frequency histograms of individual $R_0$ values (*Brugger and Althaus, 2020*). (B) Simulated and actual cumulative distribution of individual $R_0$ values. (C) Simulated and actual frequency histograms of individual serial intervals (*Cowling et al., 2009*). (D) Simulated and actual cumulative distribution of individual serial intervals. (E) Frequency distribution of simulated generation times.

good approximations of the mean and variance for generation time. A majority of generation times fell between 2 and 6 days (*Figure 6e*).

## Viral load thresholds for influenza transmission

Based on the optimized TD curve for influenza (*Figure 7a*), we next plotted the probability of infection given an exposure to an infected person. The TD50 for influenza was $10^{6.1}$ TCID/mL. Under various shedding scenarios, the window of high probability transmission was limited to time points around peak viral load (*Figure 7b–d*). In general, infected persons were likely to be most infectious (i.e., above TD50) for a ~ 0.5–1.0 day period. The observed low variance in serial interval (*Figure 6c*)

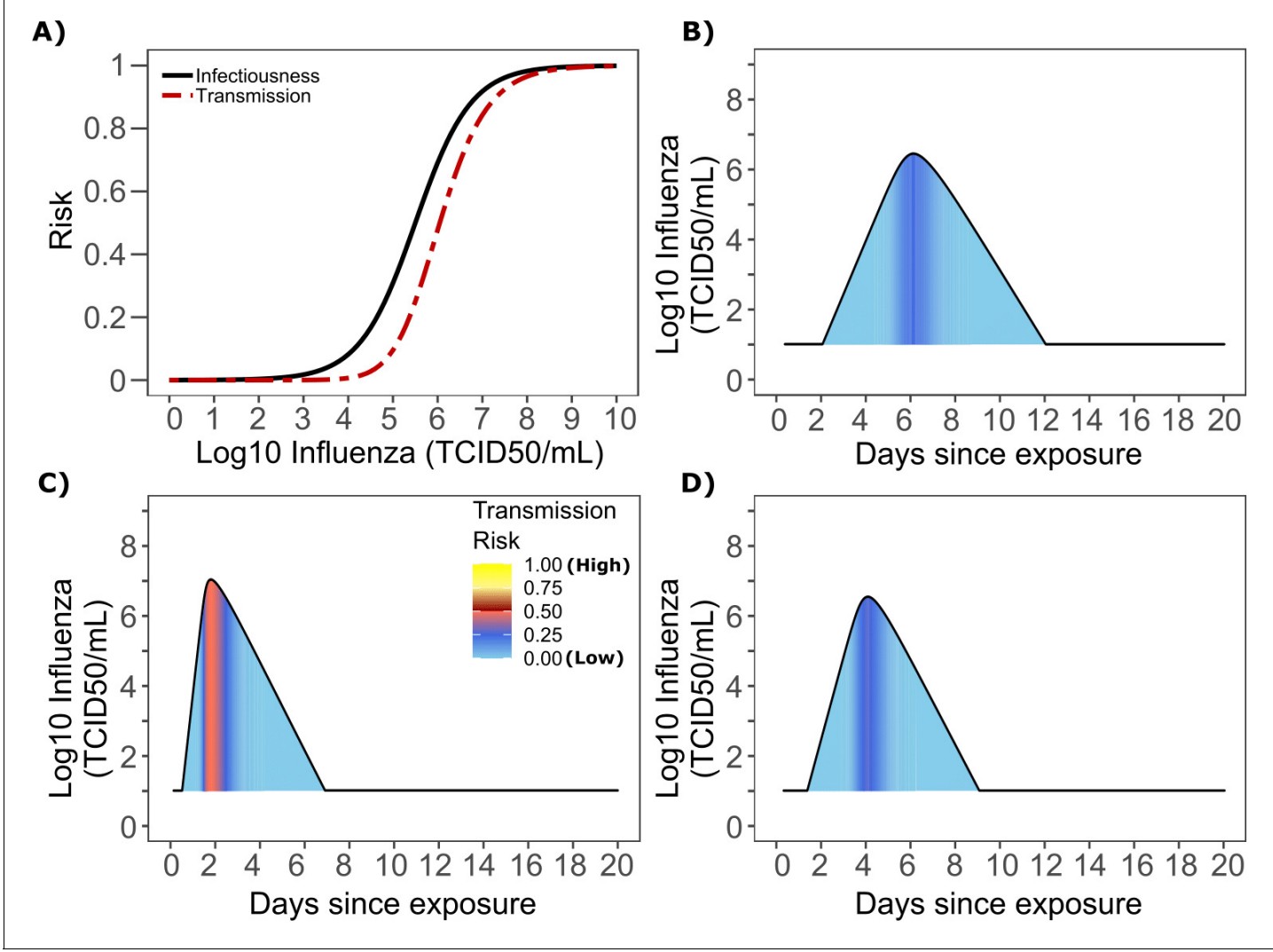

**Figure 7.** Influenza transmission probability as a function of shedding. (A) Optimal infectious dose (ID) response curve (infection risk = $P_t$) and transmission dose (TD) response curve (transmission risk = $P_t * P_t$) curves for influenza. Transmission probability is a product of two probabilities, contagiousness and infectiousness (*Figure 1*). (B-D) Three simulated viral shedding curves. Heat maps represent risk of transmission at each shedding timepoint given an exposed contact with an uninfected person at that time.

primarily results from the low variance in incubation period (*Figure 1—figure supplement 4b*) and the limited variability in timing of peak viral loads across transmitters.

### Determinants of influenza individual $R_0$

We generated a heat map from our TD curve to identify conditions governing influenza transmission to multiple people including viral load exceeding $10^6$ influenza TCID and a high number of exposure contacts per day. The contact network never resulted in days with more than 15 exposure contacts per day, which severely limited the possible number of transmissions from a single person relative to SARS-CoV-2 (*Figure 8a*, *Figure 1—figure supplement 3b*).

We plotted transmission events simulated on a daily basis over 30 days since infection from 10,000 transmitters according to viral load at exposure and number of exposure contacts on that day (*Figure 8b*). Secondary transmissions to fewer than five people accounted for 90% of infections (*Table 1*) and occurred with fewer than 10 daily exposure contacts and exposure viral loads

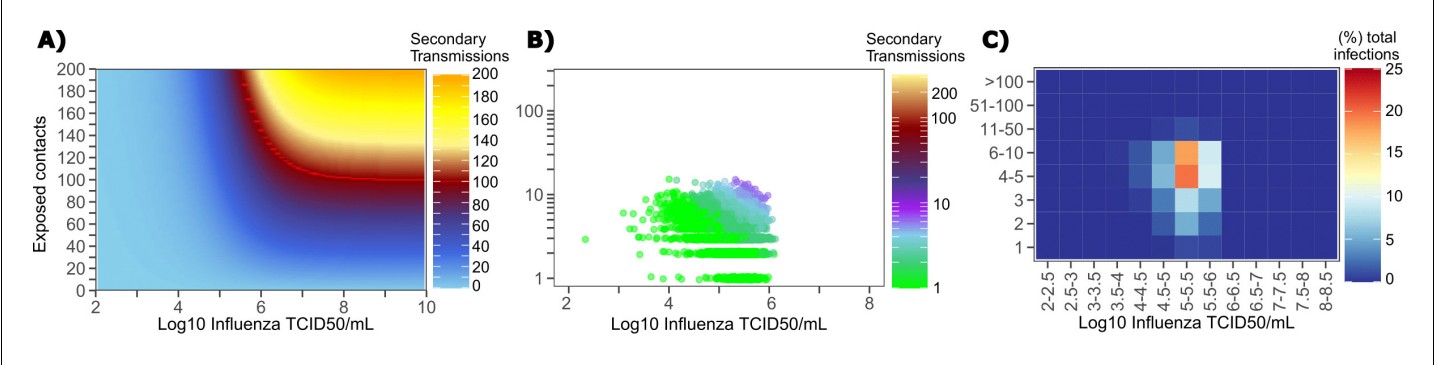

**Figure 8.** Conditional requirements for influenza super spreading events. (A) Heatmap demonstrating the maximum number of secondary infections per day feasible from a transmitter given an exposure viral load on log10 scale (x-axis) and number of exposed contacts per day (y-axis). (B) 10,000 simulated transmitters followed for 30 days. The white space is a parameter space with no transmissions. Each dot represents the number of secondary transmissions from a transmitter per day. Input variables are log10 influenza TCID on the start of that day and number of contact exposures per day for the transmitter. There are 1,239,984 total exposure contacts and 11,141 total infections. (C) 10,000 simulated infections with percent of infections due to exposure viral load binned in intervals of 0.5 intervals on log10 scale (x-axis) and number of exposed contacts (y-axis).

exceeding $10^4$ TCID. Small scale super-spreader events with 5–10 infected people almost always occurred at viral loads exceeding $10^5$ TCID with 5–10 concurrent exposure contacts (*Figure 8b*).

We next identified that over 50% of infections were associated with a transmitter who had fewer than 10 exposure contacts per day and a viral load exceeding $10^{4.5}$ TCID (*Figure 8c*), which is why no infected person ever transmitted to more than 10 other people (*Table 1*).

## Differing exposed contact distributions, rather than viral kinetics, explain SARS CoV-2 super-spreader events

We sought to explain why SARS-CoV-2 has a more over-dispersed distribution of individual $R_0$ relative to influenza. To assess viral kinetics as a potential factor, we comparatively plotted transmission risk per exposure contact as a function of time since infection in 10,000 transmitters for each virus. The median per contact transmission risk among simulated transmitters was slightly higher for influenza; however, the upper bounds of 75% and 95% percentile of transmission risks among simulated transmitters were marginally higher for SARS-CoV-2 compared to influenza along with a longer tail of low transmission risk beyond 7 days after exposure (*Figure 9a*). The transmission risk was considerably higher for the 25% of simulated SARS-CoV-2 infections with the highest viral loads, suggesting that a substantial subset of infected people may be more pre-disposed to super-spreading. When plotted as time since onset of symptoms, the variability in the time at which transmissions take place relative to symptom onset was considerably larger for persons with high SARS-CoV-2 viral load, owing to the variable incubation period of this virus (*Figure 9b*).

The median duration of shedding over infectivity thresholds was short and nearly equivalent for both viruses. For SARS-CoV-2 and influenza, median [range] time above ID10 was 2.7 [0, 7] and 2.4 [1.6, 3.7] days, respectively; median time above ID25 was 1.7 [0, 3] and 1.5 [0, 2.2] days, respectively; median time above ID50 was 0.8 [0, 1.3] and 0 [0, 1.3] days, respectively; median time above ID75 was 0 [0, 0.4] and 0 [0, 0] days, respectively; median time above ID90 was 0 [0, 0] and 0 [0, 0] days, respectively. ID10, ID25 and ID50 values were more variable across SARS-CoV-2 simulations due to more heterogeneous viral kinetics among simulated infected people.

For SARS-CoV-2 and influenza, median [range] time above TD10 was 1.4 [0, 2.5] and 1.2 [0, 2.0] days, respectively; median time above TD25 was 0.8 [0, 1.3] and 0.3 [0, 1.3] days, respectively; median time above TD50 was 0 [0, 0.5] and 0 [0, 0.4] days, respectively; median time above TD75 was 0 [0, 0] and 0 [0, 0] days, respectively. TD10, TD25 and TD50 values were more variable across SARS-CoV-2 simulations due to a minority of trajectories with prolonged moderate viral loads (*Figure 9c*).

We next plotted the frequency of exposure contacts per day for both viruses and noted a higher frequency of days with no exposed contacts (*Figure 9d*), but also a higher frequency of days with

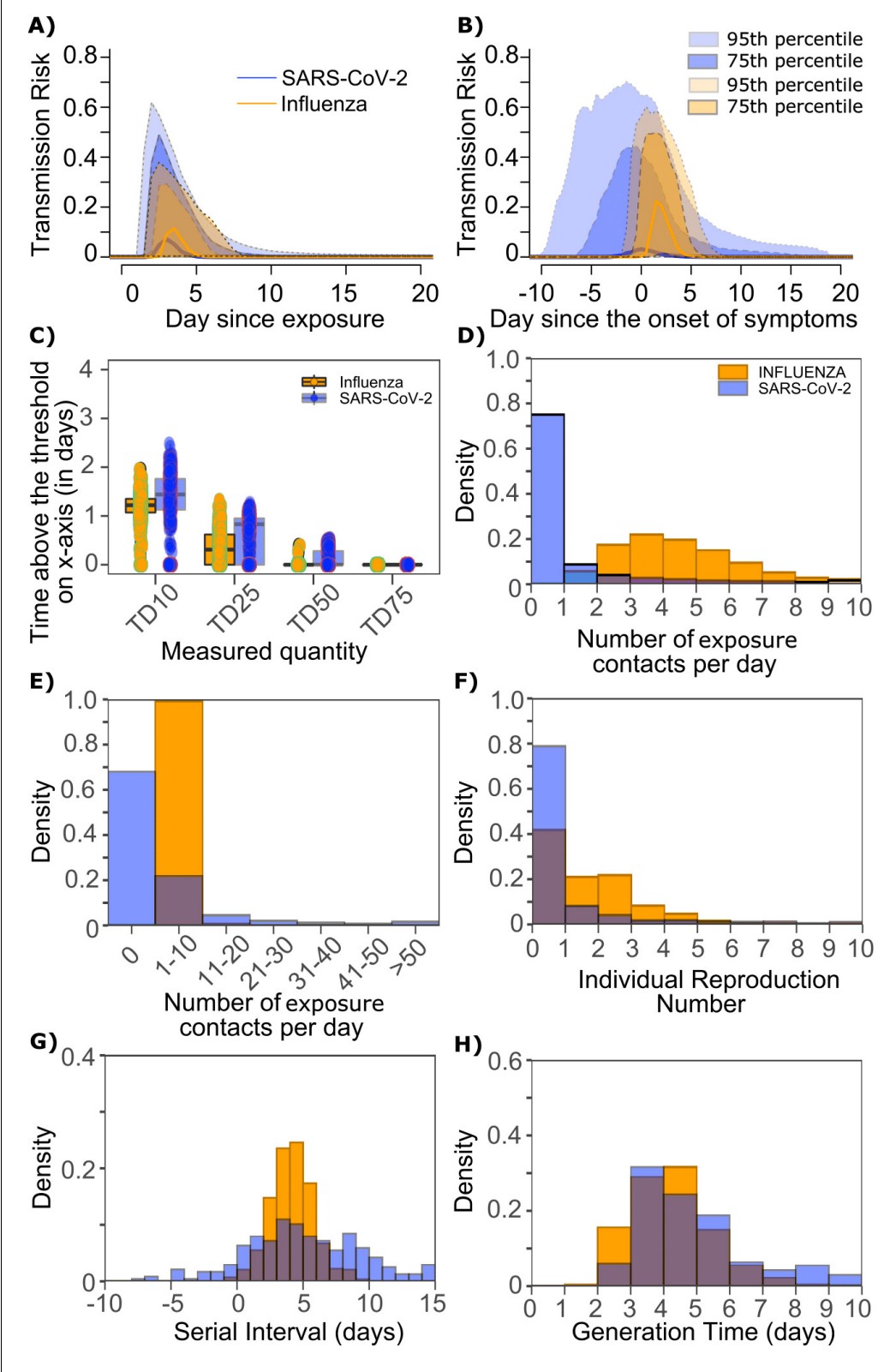

**Figure 9.** Differing transmission contact distributions, rather than viral kinetics explain SARS CoV-2 super spreader events. (**A**) Simulated transmission risk dynamics for 10,000 infected persons with SARS-CoV-2 and influenza. Solid line is median transmission risk. Dark, dotted line is transmission risk of 75th percentile viral loads, and light dotted line is transmission risk of 95th percentile viral loads. (**B**) Same as A but plotted as transmission risk since onset of symptoms. Highest transmission risk for SARS-Co-V-2 is pre-symptoms and for influenza is post symptoms. (**C**) Boxplots of duration of time

*Figure 9 continued on next page*

*Figure 9 continued*

spent above TD10, TD25, TD50, TD75, and TD90 for 10,000 simulated SARS-CoV-2 and influenza shedding episodes. TD10, TD25, TD50, TD75, and TD90 are viral loads at which transmission probability is 10%, 25%, 50%, 75%, and 90%, respectively. The midlines are median values, boxes are interquartile ranges (IQR), and datapoints are outliers. Superimposed probability distributions of: (D and E) number of exposure contacts per day, (F) individual $R_0$, (G) serial interval and (H) generation time for influenza and SARS-CoV-2.

The online version of this article includes the following figure supplement(s) for figure 9:

**Figure supplement 1.** Impact of changes in contact network heterogeneity on individual $R_0$, serial interval, and generation time.

more than 10 exposure contacts (*Figure 9e*) for SARS-CoV-2 relative to influenza, despite an equivalent mean number of daily exposure contacts. To confirm that this distribution drives the different observed distributions of individual $R_0$ values (*Figure 9f*), we simulated SARS-CoV-2 infection with an assumed $\rho$=1 (low dispersion of exposure contacts) and generated a distribution of individual $R_0$ similar to that of influenza (*Figure 9—figure supplement 1a*). Similarly, we simulated influenza infection with an assumed $\rho$=40 (high dispersion of exposure contacts) and generated a distribution of individual $R_0$ similar to that of SARS-CoV-2 (*Figure 9—figure supplement 1b.*). Under all scenarios, predicted distributions of serial interval (*Figure 9g*, *Figure 9—figure supplement 1*) and generation time (*Figure 9h*, *Figure 9—figure supplement 1*) were unchanged by shifts in the exposed contact network.

In summary, we conclude that despite differing viral shedding kinetics (*Figure 10a*, top), the kinetics of infectivity are extremely similar between influenza and SARS-CoV-2 (*Figure 10a*, bottom).

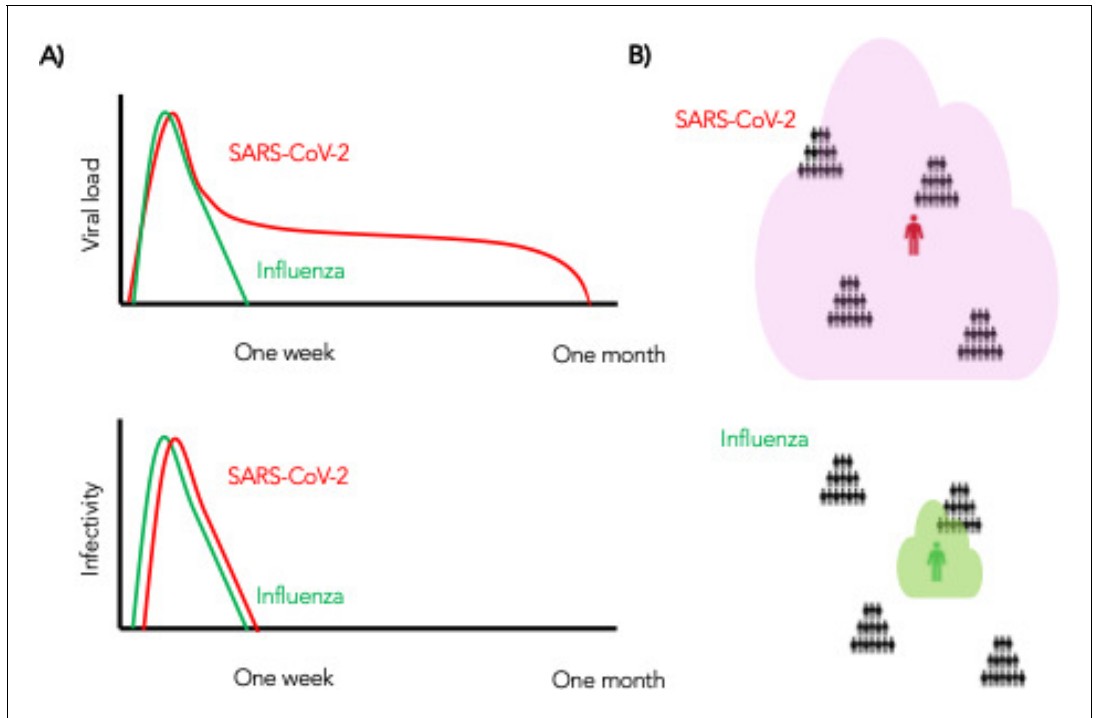

**Figure 10.** Wider dispersion of virus is the most likely explanation for SARS-CoV-2 super-spreader events. (A) Despite differing shedding kinetics, our model projects very similar kinetics of infectivity between influenza and SARS-CoV-2 (*Figure 9a,c*) but (B) higher number of exposure contacts based on wider and / or more prolonged dispersal of virus creating potentialsws for super-spreader events.

The online version of this article includes the following figure supplement(s) for figure 10:

**Figure supplement 1.** Potential impact of population physical distancing on SARS-Co-V2 epidemiology.

**Figure supplement 2.** Potential impact of enhanced physical distancing only within high exposure contact networks on SARS-CoV-2 epidemiology.

**Figure supplement 3.** Sensitivity analysis of transmission curve parameter for model fit to SARS-CoV-2 data.

**Figure supplement 4.** Sensitivity analysis of contact network structure for model fit to SARS-CoV-2 data.

**Figure supplement 5.** Histograms of four estimated parameters ($\alpha$, $\lambda$, $\theta$, and $\rho$) using Approximate Bayesian Computation rejection sampling method.

Over-dispersed contact networks, likely due to wider airborne dispersal of virus in equivalent crowded environments, explain the ability of SARS-CoV-2 to initiate super-spreader events (*Figure 10b*).

### Projections of targeted physical distancing

Physical distancing is a strategy to decrease $R_0$. We simulated a decrease in the contact rate uniformly across the population and noted a decrease in population $R_0$ (*Figure 10—figure supplement 1a*) as well the percent of infected people who will transmit (*Figure 10—figure supplement 1b*) and become super-spreaders (*Figure 10—figure supplement 1c–d*). An approximately 40% decrease in the average exposed contact rate lowered $R_0$ below 1 (*Figure 10—figure supplement 1a*). We further investigated whether lowering contact rate among larger groups only, in particular by banning exposure events with a high number of exposure contacts, could control the epidemic. We identified that limiting exposure contacts to no more than five per day is nearly equivalent to limiting exposure contacts altogether and that only a small decrease in mean exposure contact rate can achieve $R_0 < 1$ if exposure events with more than 20 contacts are eliminated (*Figure 10—figure supplement 2*).

### Pre-symptomatic transmission and super-spreading risk

Much of the highest transmission risk for SARS-CoV-2 exists in the pre-symptomatic phase (*Figure 9b*) which explains why 62% of simulated transmissions occurred in the pre-symptomatic phase for SARS-CoV-2, compared to 10% for influenza. Similarly, 62% and 21% of SARS-CoV-2 and influenza super-spreader events with secondary transmissions $\geq 5$, and 39% of SARS-CoV-2 super-spreader events with secondary transmissions $\geq 10$ fell exclusively in the pre-symptomatic period.

## Discussion

Our model provides a plausible link between SARS-CoV-2 shedding kinetics and the virus' most fundamental epidemiologic properties. First, we identify a transmission dose response curve which specifies that a nasal viral load below a certain threshold (conservatively ~$10^4$ RNA copies) is unlikely to result in transmission – consistent with the overall rarity of positive cultures at these levels (*van Kampen, 2020*). We also predict a relatively steep TD curve such that transmission becomes much more likely when a susceptible person contacts an infected person shedding above $10^8$ viral RNA copies. The amount of viral RNA can be roughly converted to the probability of a positive viral culture which approximates infectiousness, and this simulated relationship qualitatively matches the dose response observed in formal dose challenge experiments performed with SARS-CoV-1 in mice (*Watanabe et al., 2010*).

Our results may have relevance for dosing of SARS-CoV-2 in human challenge experiments that are being conducted for testing vaccines and therapies. However, we emphasize that it would first be valuable to test the model's predictions with graded challenge models of infection in non-human primates or golden hamsters (*Chandrashekar et al., 2020*). Our estimates for viral load transmission thresholds are inherently imprecise based on the fact that there is no international standard for PCR or clinical sampling site (saliva versus nasopharyngeal swab). Indeed, viral loads have varied considerably across studies and average transmission dose may have increased as social distancing measures were implemented across the globe, because higher viral load may be required to achieve transmission across larger distances and over shorter periods of time.

While the duration of shedding for SARS-CoV-2 is often three weeks or longer, (*Qi et al., 2020*; *Cao et al., 2020*) our model predicts a short period—averaging less than two days—of high transmission risk. We note that our model was fit to pandemic settings where self-isolation upon development of symptoms was normative, and this high-risk window increases to 2–3 days when fit to pre-lockdown conditions in Wuhan. This duration of infectivity is predicted to be comparable to that of influenza. Our model predicts that transmission after the first week of infection is quite rare. Output is also consistent with the observation that transmissions commonly but not always occurs during the pre-symptomatic phase of infection (*He et al., 2020*; *Moghadas et al., 2020*; *Tindale et al., 2020*). This variability is attributed less to timing of peak viral load and more to the variable incubation period of the virus.

The observed high heterogeneity in serial interval is attributable almost entirely to the variable nature of the incubation period, rather than transmission occurring extremely late after infection.

While our estimate for mean generation time is equivalent to that of mean serial interval, it is notable that the range of SARS-CoV-2 serial intervals is much wider than the range of generation times. This result is evident even though we extended heterogeneity of viral shedding curves beyond that observed in the somewhat limited existing shedding data.

The finding of limited duration of SARS-CoV-2 infectivity has practical implications. First, considerable resources are being used in hospitals and skilled nursing facilities to isolate patients with persistent SARS-CoV-2 shedding. Our model suggests that a low nasal viral load, particularly during late infection, may not require full patient isolation procedures in the absence of aerosolizing procedures. Experimental verification would be required before the implementation of such policies. If true, substantial hospital resources and valuable isolation beds might be gained for subsequent waves of infection. Similarly, employees would be able to return to work sooner after infection, saving lost labor and wages. Our results also suggest that time since first positive test may be predictive of lack of contagion, although more viral load kinetic studies will be needed to confirm the existing observation that viral loads after a week of infection are usually low and associated with negative viral cultures (*van Kampen, 2020*). Finally, our conclusions are generally supportive of rapid, less sensitive assays which are more likely to detect infection at periods of contagion (*Larremore, 2020*).

Many of these conclusions, including specific viral load thresholds for transmission, a steep dose response curve and a maximum 2 day duration of high contagion within an infected individual are equally relevant for influenza infection. One important difference is that incubation periods for influenza are far less variable, which means that at the individual level the serial interval is much more likely to be predictive of the generation time.

Another finding is that SARS-CoV-2 super-spreading events are dependent on a large number of exposure contacts during the relatively narrow 1–2 days peak infectivity window. Because we predict that super-spreader potential may be a signature property of infection, rather than a characteristic of a tiny subset of infected people, this result also has practical implications. A common experience during the pandemic has been early identification of a small cluster of infected people within a specific confined environment such as a senior living home, crowded work environment, athletic team, or restaurant. Our results demonstrate that newly diagnosed people within small clusters may be past the peak of their super-spreading potential. At this stage, many more infections have often been established and drastic quarantine procedures should be considered. Other undiagnosed, presymptomatic infected people may have super-spreader potential, while the known infected person is no longer contagious, highlighting the importance of effective contact tracing.

At the prevention level, school opening and work opening strategies should focus on severely limiting the possible number of exposure contacts per day. Where large numbers of exposure contacts are unavoidable, rigorous masking policies should be considered, perhaps with N95 masks that may more significantly lower exposure viral loads (*Leung et al., 2020*).

Influenza infection is much less predisposed to super-spreader events than SARS-CoV-2. Yet, influenza shedding at levels above those required for a high probability of transmission occurs with only slightly lower frequency. Therefore, viral kinetics are unlikely to drive the markedly different probability of super-spreader events between the two viruses—despite the fact that the overall duration of SARS-CoV-2 shedding exceeds duration of influenza shedding often by more than 2 weeks.

Rather, our analysis suggests that the exposure contact networks of SARS-CoV-2 transmitters are highly dispersed relative to those of influenza. This observation is unlikely to relate to different societal contact matrices as both viruses share the respiratory transmission route, with demonstrated ability to spread in schools, homes, workplaces. and other crowded environments. The most likely explanation underlying differing exposure contact networks is that SARS-CoV-2 is more predisposed to airborne transmission than influenza (*van Doremalen et al., 2020*). Here, our precise definition of an exposure contact (sufficient contact between a transmitter and an uninfected person to potentially allow transmission) is of high relevance. Our result suggests that a SARS-CoV-2-infected person in a crowded poorly ventilated room will generate more exposure contacts than an influenza infected person in the exact same room, likely based on wider dispersal and / or longer airborne survival of the virus. Thus, airborne transmission of SARS-CoV-2 has a significant downstream effect on epidemiology. This prediction reinforces current public health recommendation to avoid crowded indoor spaces with poor air recirculation.

On the other hand, a much higher proportion of SARS-CoV-2-infected people than influenza infected people do not transmit at all. This result lacks a clear mechanistic explanation but may imply

that aerosolization occurs only in a subset of infected people. One theoretical explanation is that high viral load shedding in the pre-symptomatic phase is defined by lack of cough or sneeze leading to limited spatial diffusion of virus. Alternatively, it is also possible that a proportion of infected people never shed virus at high enough viral loads to allow efficient transmission. This possibility speaks to the need for more quantitative viral load data gathered during the initial stages of infection (**Kissler, 2020**).

Age cohort structure differs between the two infections, with a lower proportion of observed pediatric infections for SARS-CoV-2. If adults have more high exposure events than children, then this could also explain super-spreader events. We are less enthusiastic about this hypothesis. First, SARS-CoV-2 super-spreader events have occurred in schools and camps and would likely be more common in the absence of widespread global school closures in high prevalence regions. Second, a sufficient proportion of influenza cases occur in adults to rule out the presence of frequent large super-spreading events in this population.

Our analysis has important limitations. First, exposure contacts were assumed to be homogeneous and we do not capture the volume of the exposing aerosol or droplet. If a large-volume droplet contains 10 times more viral particles than an aerosol droplet, then the exposure could be dictated by this volume as well as the viral load of the potential transmitter. It is possible that under rare circumstances with extremely high-volume exposures, even persons with extremely low viral loads may transmit.

Second, based on the quality of available data, we fit our models for SARS-CoV-2 and influenza to viral RNA and viral culture, respectively. This might impact the quantitative results but is unlikely to affect qualitative predictions of the model. Existing data suggest that kinetics of viral RNA and culture are similar during both infections, with culture having lower sensitivity to detect virus (**van Kampen, 2020**).

Third, data during early SARS-CoV-2 infection is fairly limited such that we may be underestimating variance of initial viral growth rates. We accounted for this by imputing additional heterogeneity in viral kinetic characteristics and obtained similar results regarding mechanisms of transmission. Nevertheless, it is possible that viral shedding kinetics among infected people may be more variable than observed to date. Importantly, we cannot rule out the possibility that a small minority of infected people shed at sufficient levels for transmission for much longer than has been observed to date. We also assume that viral load does not change substantially over the course of the subsequent day when simulating transmission which may allow for slight misclassification of exposure viral load.

Fourth, our intra-host model of SARS-CoV-2 was fit to heterogeneous data from different continents and with different sampling techniques and PCR assays (**Goyal et al., 2020**). Similarly, $R_0$ estimates for SARS-CoV-2 have varied temporally and spatially across the globe and the same is likely true for influenza. Social interaction matrices are very likely to differ among countries and between urban and rural areas. For these reasons, our estimates of TD50 are necessarily imprecise based on available data and should serve only as a conservative benchmark. Regarding overdispersion of individual $R_0$, we do not capture the fact that social interaction networks in certain municipalities may predispose to more frequent and severe super-spreader events. Nevertheless, we are confident that differing aerosolization properties of the two viruses underly the observed overdispersion of individual $R_0$ for SARS-CoV-2 relative to influenza, regardless of geographic location.

Fifth, contagiousness could have different dose response dynamics than viral load dependent infectiousness and may require investigation in the future upon the availability of epidemiologically relevant additional data.

Overall, the model is intended to capture a universal property of SARS-CoV-2 infection but is not specific for local epidemics. Nevertheless, it is extremely clear that super-spreader events are a globally generalizable feature of SARS-CoV-2 epidemiology.

In conclusion, fundamental epidemiologic features of SARS-CoV-2 and influenza infections can be directly related to viral shedding patterns in the upper airway as well as the nature of exposure contact networks. We contend that this information should be leveraged for more nuanced public health practice in the next phase of the pandemic.

## Materials and methods

### SARS-CoV-2 within-host model

To simulate SARS-CoV-2 shedding dynamics, we employed our previously described viral infection model (*Goyal et al., 2020*). In this model, susceptible cells ($S$) after coming into contact with SARS-CoV-2 ($V$) become infected at rate $\beta VS$. The infected cells ($I$) produce new virus at a per-capita rate $\pi$. The model also includes the clearance of infected cells in two ways: (1) by an innate response with density dependent rate $\delta I^k$; and (2) an acquired response with rate $\frac{mE^r}{E^r + \phi^r}$ mediated by SARS-CoV-2-specific effector cells ($E$). The clearance mediated by innate immunity depends on the infected cell density and is controlled by the exponent $k$. The Hill coefficient $r$ parameterizes the nonlinearity of the second response and allows for rapid saturation of the killing. Parameter $\phi$ defines the effector cell level by which killing of infected cells by $E$ is half maximal.

In the model, SARS-CoV-2-specific effector cells rise after two stages from precursors cells ($M_1$ and $M_2$). The first precursor cell compartment ($M_1$) proliferates in the presence of infection with rate $\omega I M_1$ and differentiates into the effector cell at a per capita rate $q$ during the next intermediate stage. Finally, effector cells die at rate $\delta_E$. The model is expressed as a system of ordinary differential equations:

$$\begin{aligned}
\frac{dS}{dt} &= -\beta VS \\
\frac{dI}{dt} &= \beta VS - \delta I^k I - m \frac{E^r}{E^r + \phi^r} I \\
\frac{dV}{dt} &= \pi I - \gamma V \\
\frac{dM_1}{dt} &= \omega I M_1 - q M_1 \\
\frac{dM_2}{dt} &= q(M_1 - M_2) \\
\frac{dE}{dt} &= q M_2 - \delta_E E
\end{aligned}$$

We assumed $S(0) = 10^7$ cells/mL, $I(0) = 1$ cells/mL, $V(0) = \frac{\pi I(0)}{c}$ copies/mL, $M_1(0) = 1$, $M_2(0) = 0$ and $E_0 = 0$.

When we introduce simulated heterogeneity in cases of SARS-CoV-2 (by increasing the standard deviation of the random effects of parameters $\beta$ by 20, $\delta$ by 2, $k$ by 2, and $\pi$ by 5 in the original distribution from *Goyal et al., 2020*), some of the viral shedding curves suggest that viral shedding could continue for long period (over 6 weeks). Indeed, while median viral shedding duration has been estimated at 12–20 days, shedding for many months is also observed commonly (*Widders et al., 2020*). We assumed that viral loads after day 20 drop to a exposure-level viral load level (i.e. $V(0)$) as most viral shedding observed after this point is transient and at an extremely low viral load (*Huang, 2020*). The population distribution of parameters to simulate artificial SARS-CoV-2 viral shedding dynamics is provided in *Table 2*.

### Influenza within-host model

To simulate viral shedding dynamics of influenza viral, we employ a model (*Baccam et al., 2006*) that is a simplified version of the viral dynamics model presented for SARS-CoV-2. This model assumes $k = 0$ and $m = 0$ and can be expressed as a system of ordinary differential equations:

**Table 2.** Population parameter estimates for simulated SARS-CoV-2 viral shedding dynamics.
Parameters are from (doi: https://doi.org/10.1101/2020.04.10.20061325). The top row is the fixed effects (mean) and the bottom row is the standard deviation of the random effects. We also fixed r = 10, δE = 1/day, q = 2.4 × 10–5/day, and c = 15/day.

| $Log_{10}\beta$ (virions$^{-1}$ day$^{-1}$) | $\delta$ (day$^{-1}$ cells$^{-k}$) | k (-) | $Log_{10}\pi$ (log$_{10}$ day$^{-1}$) | m (day$^{-1}$ cells$^{-1}$) | $Log_{10}\omega$ (day$^{-1}$ cells$^{-1}$) |
|---|---|---|---|---|---|
| −7.23 | 3.13 | 0.08 | 2.59 | 3.21 | −4.55 |
| 0.2 | 0.02 | 0.02 | 0.05 | 0.33 | 0.01 |

$$\frac{dS}{dt} = -\beta V S$$

$$\frac{dI}{dt} = \beta V S - \delta I$$

$$\frac{dV}{dt} = \pi I - \gamma V$$

Following this model, (*Baccam et al., 2006*) we assumed $S(0) = 4 \times 10^8$ cells/mL, $I(0) = 1$ cells/mL, $V(0) = \frac{\pi I(0)}{c}$ copies/mL. To simulate artificial influenza viral shedding dynamics, we assumed the population distribution of parameters $Log10(\beta)$, $Log10(\pi)$, $Log10(\gamma)$ and $Log10(\delta)$ are -4.56 (0.17), -1.98 (0.14), 0.47 (0.03), and 0.60 (0.06), respectively.

## Dose-response model

For both viruses, to estimate the infectiousness $P_t[V(t)]$ based on viral loads $V(t)$, we employed the function, $P_t[V(t)] = \frac{V(t)^\alpha}{\lambda^\alpha + V(t)^\alpha}$. Here, $\lambda$ is the infectivity parameter that represents the viral load that corresponds to 50% infectiousness and 50% contagiousness, and $\alpha$ is the Hill coefficient that controls the slope of the dose-response curve.

## Transmission model and reproduction number

Our transmission model assumes that only some contacts of an infected individual with viral load dependent infectiousness are physically exposed to the virus (defined as exposure contacts), that only some exposure contacts have virus passaged to their airways (contagiousness) and that only some exposed contacts with virus in their airways become secondarily infected (successful secondary infection). Contagiousness and infectiousness are then treated as viral load dependent multiplicative probabilities with transmission risk for a single exposure contact being the product. Contagiousness is considered to be viral load dependent based on the concept that a transmitter's dispersal cloud of virus is more likely to prove contagious at higher viral load, which is entirely separate from viral infectivity within the airway once a virus contacts the surface of susceptible cells.

We next assume that the total exposed contacts within a time step $\left(\eta_{\Delta_t}\right)$ is gamma distributed, that is, $\eta_{\Delta_t} \sim \Gamma\left(\frac{\theta}{\rho}, \rho\right)\Delta_t$, using the average daily contact rates $(\theta)$ and the dispersion parameter $(\rho)$. To obtain the true number of exposure contacts with airway exposure to virus, we simply multiply the contagiousness of the transmitter with the total exposed contacts within a time step (i.e., $\zeta_t = \eta_{\Delta_t} P_t$).

Transmissions within a time step are simulated stochastically using time-dependent viral load to determine infectiousness $(P_t)$. Successful transmission is modeled stochastically by drawing a random uniform variable $(U(0,1))$ and comparing it with infectiousness of the transmitter. In the case of successful transmission, the number of secondary infections within that time step $(T_{\Delta_t})$ is obtained by the product of the infectiousness $(P_t)$ and the number of exposure contacts drawn from the gamma distribution $(\zeta_t)$. In other words, the number of secondary infections for a time step is $T_{\Delta_t} = Ber(P_t)P_t\eta_{\Delta_t}$. If we disregard contagiousness by assuming $P_t = 1$ in $\zeta_t$, we identify that there are little to no differences on overall results other than the emergent TD curve and optimal parameter set describing dose-response curve and exposed contact network, which no longer agrees as closely with in vitro probability of positive virus culture (*Figure 2—figure supplement 1*; *van Kampen, 2020*).

We obtain the number of secondary infections from a transmitter on a daily basis assuming that viral load, and subsequent risk, does not change substantially within a day. We then summed up the number of secondary infections over 30 days since the time of exposure to obtain the individual reproduction number, that is, $R_0 = \sum_{\Delta_t} T_{\Delta_t}$.

## Serial interval and generation time

We further assume that upon successful infection, it takes $\tau$ days for the virus to move within-host, reach infection site and produce the first infected cell.

To calculate serial interval (time between the onset of symptoms of transmitter and secondarily infected person), we sample the incubation period in the transmitter and in the secondarily infected person from a gamma distribution with a shape described in *Figure 1—figure supplement 4* (*Ganyani et al., 2020*; *Lauer et al., 2020*). In cases in which symptom onset in the newly infected person precedes symptom onset in the transmitter, the serial interval is negative; otherwise, serial interval is non-negative. We calculate generation time as the difference between the time of infection of transmitter and the time of infection of secondarily infected person.

## Individual $R_0$ and serial interval data for model fitting

There is abundance of data confirming over-dispersed $R_0$ for SARS-CoV-2. From contact tracing of 391 SARS-CoV-2 cases in Shenzhen, China, 1286 close contacts were identified: the distribution of individual $R_0$ values in this cohort was highly over-dispersed, with 80% of secondary infections being caused by 8–9% of infected people (*Bi et al., 2020*). In another study, authors analyzed the contact/travel history of 135 infected cases in Tianjin, China and determined heterogeneity in the individual $R_0$. (*Zhang et al., 2020*) Another contract tracing study also identified and characterized SARS-CoV-2 clusters in Hong Kong and estimated that 20% of cases were responsible for 80% of local transmission (*Dillon, 2020*).

A modeling study that simulated observed outbreak sizes in ~40 affected countries during the early phase of epidemics also confirmed that ~80% of secondary transmissions may have been caused by a small fraction of infectious individuals (~10%) (*Endo et al., 2020*). The latter study provided the distribution of individual $R_0$ (*Figure 2a*) that we employed for fitting purposes. Using the data on 468 COVID-19 transmission events reported in mainland China, Du et al. estimated the mean serial interval as well as the distribution of serial interval (*Figure 2c*). (*Du et al., 2020*) We employed this data for fitting purposes. Alternatively, we also fit to the mean serial interval as well as the distribution of serial interval formulated from data of 162 transmission pairs that were observed before January 22, 2020 (termed as pre-lockdown) during the initial stages of the pandemic in Wuhan (*Ganyani et al., 2020*).

The cumulative distribution function of individual $R_0$ for influenza was obtained from a modeling study that simulated the transmission dynamics of seasonal influenza in Switzerland from 2003 to 2015 (*Brugger and Althaus, 2020*). We picked the parameters mean $R_0$ = 1.26 and dispersion parameter = 2.36 in the negative binomial distribution that corresponded to the 2008–2009 influenza A H1N1 epidemic season (*Brugger and Althaus, 2020*). Another modeling study that simulated the age-specific cumulative incidence of 2009 H1N1 influenza in eight Southern Hemisphere Countries yielded similar results (*Opatowski et al., 2011*). By following the household members of index cases, a study estimated the cumulative distribution of serial interval based on symptom-onset times from 14 transmission pairs (*Cowling et al., 2009*). We employed these cumulative distribution functions of individual $R_0$ and serial interval of influenza for fitting purposes.

## Fitting procedure

To estimate the values of unknown parameters in cases of SARS-CoV-2, we performed a grid search comprehensively exploring a total of 417,792 combinations of 5 parameters taking the following values:

i.    $\tau \in$ [0.5, 1, 2, 3] days,
ii.   $\alpha \in$ [0.01, 0.1, 0.2, 0.3, 0.4, 0.5, 0.6, 0.7, 0.8, 0.9, 1.0, 2.0, 3.0, 4.0, 5.0, 10.0]
iii.  $\lambda \in [10^0, 10^{0.5}, 10^{1.0} \ldots, 10^8]$
iv.   $\theta \in$ [0.1, 0.2, 0.3, 0.4, 0.5, 0.6, 0.7, 0.8, 0.9, 1.0, 2.0, 3.0, 4.0, 5.0, 10.0, 20.0, 50.0].
v.    $\rho \in$ [0.0001, 0.001, 0.01, 0.1, 0.2, 0.3, 0.4, 0.5, 0.6, 0.7, 0.8, 0.9, 1.0, 2.0, 5.0, 10.0, 20.0, 30.0, 40.0, 50.0, 75.0, 100, 200, 500].

The parameter sets of ($\lambda$, $\tau$, $\alpha$, $\theta$, $\rho$) were simulated for 1000 infected individuals to determine how well each set generates the summary statistics of mean $R_0$, mean SI and the $R_0$ histograms by following a procedure explained in *Figure 1—figure supplement 1* and below:

## Step A

1. Simulate viral load $V(t)$ of 1000 simulated infected individuals using the differential equation for virus described for SARS-CoV-2 and influenza above:

2. For each combination of ($\lambda$, $\tau$, $\alpha$, $\theta$, $\rho$)
   a. For each time step $\Delta_t$
      i. Compute $P_t[V(t); \lambda, \alpha]$
      ii. Draw $\eta_{\Delta_t} \sim \Gamma\left(\frac{\theta}{\rho}, \rho\right)\Delta_t$
      iii. Calculate $T_{\Delta_t} = Ber(P_t)P_t\eta_{\Delta_t}$
   b. Calculate $R_0 = \sum_{\Delta_t} T_{\Delta_t}$
      i. Check if calculated mean $R_0$ is in the range: (*Ganyani et al., 2020*; *Du et al., 2020*)
   c. Calculate Serial Interval based on $\tau$ and incubation period
      i. Check if calculated *SI* is in the range in: (*Ganyani et al., 2020*; *Du et al., 2020*; *Nishiura et al., 2020*)

## Step B

1. If the parameter combination in Step A satisfy the criteria, then
   i. Compute residual sum of squares (RSS) for the obtained $R_0$ and histogram from: (*Endo et al., 2020*; *Bi et al., 2020*; *Zhang et al., 2020*; *Miller, 2020*)

We visually checked whether our dose-response curve matched the observed probability of positive virus culture (*van Kampen, 2020*). We assumed that viral loads derived from positive culture (*van Kampen, 2020*) can be considered equivalent to viral loads in the within-host model if divided by a positive integer. We identified an integer of 25 to provide closest fit to the empirical data (*Figure 2—figure supplement 1*).

We performed a global sensitivity analysis to identify which parameter variability accounted for fit to different components of the data. Only narrow ranges of λ permitted close fit to the mean of R₀ and distribution functions of individual R₀ (*Figure 10—figure supplement 3*), while a specific value for α was necessary to fit to mean serial interval and distribution functions of individual R₀ (*Figure 10—figure supplement 3*). Only narrow ranges of θ permitted close fit to the mean of R₀ and distribution functions of individual R₀ (*Figure 10—figure supplement 4*), while a specific value for ρ was necessary to fit to distribution functions of individual R₀ (*Figure 10—figure supplement 4*).

To obtain TD50 ($\lambda_T$) based on ID50 ($\lambda$), we use the following relation for real positive values of α,

$$\frac{1}{\left(\left(\frac{10^\lambda}{V}\right)^\alpha + 1\right)^2} = \frac{1}{\left(\frac{10^{\lambda_T}}{V}\right)^{\alpha_T} + 1} = 0.5$$

From solving the second half ($\frac{1}{\left(\frac{10^{\lambda_T}}{V}\right)^{\alpha_T} + 1} = 0.5$), we get

$$V = 10^{\lambda_T}$$

Substituting $V = 10^{\lambda_T}$ in the first-half, we have,

$$\frac{1}{\left(\left(\frac{10^\lambda}{10^{\lambda_T}}\right)^\alpha + 1\right)^2} = 0.5$$

$$\text{Or, } \left(\left(\frac{10^\lambda}{10^{\lambda_T}}\right)^\alpha + 1\right)^2 = 2$$

$$\text{Or, } \left(\frac{10^\lambda}{10^{\lambda_T}}\right)^\alpha = \sqrt{2} - 1$$

$$\text{Or, } 10^{\lambda_T\alpha} = \frac{10^{\lambda\alpha}}{\sqrt{2} - 1}$$

$$\text{Or, } \lambda_T = \lambda + \frac{0.38}{\alpha}$$

## Alternative fitting procedures

To further confirm the validity and robustness of our results, particularly in relation to the estimated parameter values, we alternatively employed Approximate Bayesian Computation (ABC) rejection-sampling method to estimate 1000 combinations of four parameters ($\alpha$, $\lambda$, $\theta$, $\rho$) with an error threshold of 0.1 while assuming $\tau = 0.5$ days. We assumed uniform prior distribution of each parameter with ranges described in the previous section. This approach yields a parameter distribution for all parameters except the parameter $\alpha$ that is centered at parameter values generated by the grid search method (*Figure 10—figure supplement 5*). The value for the parameter $\alpha$ can be further narrowed by the use of additional fitting to the data of the probability of positive virus culture in vitro.

Furthermore, we also employed a narrow grid search close to the solution yielded by both the extensive grid search in the previous section and ABC. In this procedure, we searched a total of ~8000 parameter combinations varying, $\alpha$ from 0.5 to 0.1 at intervals of 0.05; $\lambda$ from 6.5 to 7.5 at intervals of 0.1; $\theta$ from 3 to 5 at intervals of 0.2, and $\rho$ from 30 to 50 at intervals of 5 while keeping $\tau$ fixed at 0.5 days. This process yielded most likely parameter estimates (with the lowest error threshold of 0.02) with median $\lambda$, $\theta$, $\rho$ and $\alpha$ taking values 6.9, 3.8, 45, and 0.9, respectively. We have not visually represented the results as the figure looks very similar to *Figure 10—figure supplement 3* and *Figure 10—figure supplement 4* but on a narrow range of parameter values.

Overall, these two alternative approaches confirm that our parameter estimates obtained for our model using the grid search method in the previous section are the most likely parameter values given the features of the COVID-19 pandemic.

## Data and materials availability

The original data and code is shared at: https://github.com/ashish2goyal/SARS_CoV_2_Super_Spreader_Event; *Goyal, 2021*; copy archived at swh:1:rev:1b445c424f077d248da4860e56825e82a5307eb6.

## Acknowledgements

We are grateful to study participants from around the globe who donated critical virologic data early during the pandemic. We thank Jeroen van Kampen and Marion Koopmans for helpful discussions. Funding: This study was supported by Fred Hutchinson Cancer Research Center faculty discretionary funds and by National Institute of Allergy and Infectious Diseases (grant # 5R0 1AI121129-05).

## Additional information

### Competing interests

Joshua T Schiffer: Reviewing editor, *eLife*. The other authors declare that no competing interests exist.

### Funding

| Funder | Grant reference number | Author |
|---|---|---|
| National Institute of Allergy and Infectious Diseases | R01 AI121129-05S1 | Joshua T Schiffer |
| Council of State and Territorial Epidemiologists | Inform Public Health Decision Making Funding Opportunity | Joshua T Schiffer |
| Fred Hutchinson Cancer Research Center | | Joshua T Schiffer |

The funders had no role in study design, data collection and interpretation, or the decision to submit the work for publication.

## Author contributions

Ashish Goyal, Formal analysis, Investigation, Visualization, Methodology, Writing - review and editing; Daniel B Reeves, Formal analysis, Supervision, Visualization; E Fabian Cardozo-Ojeda, Conceptualization, Formal analysis, Supervision, Investigation, Methodology, Project administration, Writing - review and editing; Joshua T Schiffer, Conceptualization, Resources, Supervision, Visualization, Methodology, Writing - original draft, Project administration, Writing - review and editing; Bryan T Mayer, Conceptualization, Formal analysis, Supervision, Validation, Investigation, Visualization, Methodology, Project administration, Writing - review and editing

## Author ORCIDs

Daniel B Reeves (iD) http://orcid.org/0000-0001-5684-9538
Joshua T Schiffer (iD) https://orcid.org/0000-0002-2598-1621

## Decision letter and Author response

Decision letter https://doi.org/10.7554/eLife.63537.sa1
Author response https://doi.org/10.7554/eLife.63537.sa2

## Additional files

### Supplementary files

- Transparent reporting form

### Data availability

The original data and code is shared at: https://github.com/ashish2goyal/SARS_CoV_2_Super_Spreader_Event copy archived at https://archive.softwareheritage.org/swh:1:rev:1b445c424f077d248da4860e56825e82a5307eb6/.

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
