## [Decision Letter]

**Acceptance summary:**

This work assesses the role of within-host viral shedding dynamics and contact heterogeneity on distribution of transmission events in SARS-CoV-2 and influenza. Using multi-scale modeling, predictions are made on the manner and contribution of super spreading to transmission. This model has the potential to provide insight into transmission dynamics of SARS-CoV-2, which could help inform policy.

**Decision letter after peer review:**

Thank you for submitting your article "Viral load and contact network predict SARS-CoV-2 transmission and super-spreading events" for consideration by *eLife*. Your article has been reviewed by three peer reviewers, one of whom is a member of our Board of Reviewing Editors, and the evaluation has been overseen by Aleksandra Walczak as the Senior Editor. The following individual involved in review of your submission has agreed to reveal their identity: Jonathan Forde (Reviewer #3).

The reviewers have discussed the reviews with one another and the Reviewing Editor has drafted this decision to help you prepare a revised submission.

Summary:

This work assesses the role of within-host viral shedding dynamics and contact heterogeneity on distribution of transmission events in SARS-CoV-2 and influenza. Using multi-scale modeling, with similar resulting generation time and serial interval distributions to published work, predictions are made on the manner and contribution of super spreading to transmission. Distinctions are seen when comparing to applying a similar modeling framework to influenza.

Essential revisions:

1) Statistical analysis:

The model parameters are estimated using an exhaustive grid search, which yields good fits for the best-fit values, but there is no assessment of statistical certainty in the parameter values. The authors essentially adopted a strategy in the spirit of approximate Bayesian computation (ABC), by proposing parameter values, simulating from a model, and comparing summary statistics of the simulated output to known values from the literature. The analysis would be helped by doing a more formal ABC analysis, as this would provide a better sense of how narrowly constrained the parameter values are given the available data. At minimum, it would be more convincing to consider additional parameter sets grided across a narrowed region of parameter space before selecting an optimal fit.

2) Model validation

The state of our knowledge about these infections is limited, both by the short time during with this research has been conducted, and the paper's need to rely on data taken from before the introduction of confounding factors such as social distancing and widespread mask usage. For this reason, in addition to the included sensitivity analysis for the model parameters, a sense of the sensitivity of the model's conclusions to the data set to which it is being fitted is needed. How much would these results change if there are errors in our understanding of the distribution of individual R0 values, or serial intervals?

3) Distinction in assumptions for flu and covid

The populations on which the histograms for the two diseases are based are quite different. For SARS-CoV-2, the studies are from China (Shenzhen, Tianjin and Hong Kong), while those for influenza are from Switzerland. Could cultural differences be relevant? What about seasonal differences, as the time during which the early SARS-CoV-2 studies occurred was necessarily restricted?

Furthermore, the explanation for the difference between influenza and COVID is based primarily on differences in contact patterns. While the Discussion clarifies this to be based on the efficiency with which exposures lead to infections (and pre-symptomatic transmission), which does sound like a viral parameter, rather than a social one. These viral factors do seem more believable than having to explain why the patterns of social contact exhibited by influenza patients would differ from those of SARS-CoV-2 patients. More focus on possible mechanistic explanations is warranted.

Title: "Contact heterogeneity" rather than "contact network" is more appropriate for the title as no network is considered.

[Editors' note: further revisions were suggested prior to acceptance, as described below.]

Thank you for submitting your article "Viral load and contact heterogeneity predict SARS-CoV-2 transmission and super-spreading events" for consideration by *eLife*. Your article has been reviewed by three peer reviewers, one of whom is a member of our Board of Reviewing Editors, and the evaluation has been overseen by Aleksandra Walczak as the Senior Editor. The Reviewing Editor has drafted this decision to help you prepare a revised submission.

Revisions:

A key definition in the work (as stated in the Discussion) is the exposure contact. However, apart from the parenthetical definition in the Discussion, exposure contact is not clearly defined. This is particularly important as estimates on the exposure contact seem to impact the estimates on the viral load to infectiousness functional relationship. In particular, infectiousness is defined "… as the viral load dependent probability of transmission given direct airway exposure to virus in an exposure contact." Why does the parameterization for this change so significantly pre and post lockdown when the main difference appears to be in exposure contacts? The explanation given refers to "… more prolonged and intense exposure contacts…" This seems to imply an unequal reflection of what an "exposure contact" truly means. Furthermore, how does the "exposed contact rate" compare with exposure contacts.

Figure 10. This new figure is confusing. In the text, viral shedding kinetics are referred to as panel (A) and kinetics of infectivity as panel (B), but they are both part of panel (A) in the figure. What is different between the SARS-CoV-2 and Influenza schematics in panel (B). Panel (C) is referred to in the text but is not in the figure.

---

## [Author Response]

Essential revisions:1) Statistical analysis:The model parameters are estimated using an exhaustive grid search, which yields good fits for the best-fit values, but there is no assessment of statistical certainty in the parameter values. The authors essentially adopted a strategy in the spirit of approximate Bayesian computation (ABC), by proposing parameter values, simulating from a model, and comparing summary statistics of the simulated output to known values from the literature. The analysis would be helped by doing a more formal ABC analysis, as this would provide a better sense of how narrowly constrained the parameter values are given the available data. At minimum, it would be more convincing to consider additional parameter sets grided across a narrowed region of parameter space before selecting an optimal fit.

Thank you for this valuable feedback. We agree that a more detailed evaluation of the parameter space would be valuable. In the first version of the manuscript, we demonstrate in Figure 10—figure supplement 3 and Figure 10—figure supplement 4 that there is a fairly narrow parameter space both for the dose response curve (Figure 10—figure supplement 3) and contact network space (Figure 10—figure supplement 4) that provides best model fit. Given that we are only exploring 4 parameters, we are confident that our parameter estimates are fairly precise.

We appreciate the idea to test the model fit within a tighter parameter space, in proximity to the existing optimal solution. We performed this analysis with λ varied between 6.5 and 7.5 logs at intervals of 0.1, α varied between 0.5 and 1.0 at intervals of 0.05, θ varied between 3 and 5 at intervals of 0.2, and ρ varied between 30 and 50 at intervals of 5, with narrower intervals to allow more precise parameter estimation. The estimates, now reported in the manuscript, only change slightly with this approach.

We also performed ABC to gather 1,000 parameter combinations with an error threshold of 0.1. This approach further confirms that our previous parameter estimates are the most likely parameter values given the features of the COVID-19 pandemic (see Author response image 1). We have summarized these new results in the revised manuscript as in “The preciseness of parameter estimates was independently confirmed with the use of Approximate Bayesian Computation (ABC) rejection sampling method (https://www.nature.com/articles/npre.2011.5964.1) and a finer grid search in proximity to the aforementioned optimal solution” in the revised manuscript and in Figure 10—figure supplement 5.

**Author response image 1. sa2fig1:** ABC approach to estimate 4 unknown parameters in the model. We estimated 1000 parameter combinations of four parameters with an error threshold of 0.1 and showed them in the form of a boxplot.

2) Model validationThe state of our knowledge about these infections is limited, both by the short time during with this research has been conducted, and the paper's need to rely on data taken from before the introduction of confounding factors such as social distancing and widespread mask usage. For this reason, in addition to the included sensitivity analysis for the model parameters, a sense of the sensitivity of the model's conclusions to the data set to which it is being fitted is needed. How much would these results change if there are errors in our understanding of the distribution of individual R0 values, or serial intervals?

This is an excellent point. We now include data which demonstrates that the serial interval was longer during the initial stages of the pandemic in Wuhan than in subsequent cohorts across the globe (PMID: 31995857). This makes sense because infected people are currently much more likely to self-isolate after developing symptoms than in the very earliest phases of the pandemic. Essentially, a higher proportion of pre-lockdown transmissions occurred during the symptomatic phase compared to post-lockdown. We therefore have additionally re-fit the model to this data and show that the duration of infectivity is predicted to be significantly longer under this assumption. This is now included as Figure 5 in the revised version.

We similarly fit the model to presumed data with higher R0 of 2.8-2.9 (see Author response image 2, which demonstrates excellent fits to the individual R0 distribution drawn from the negative binomial distribution with assumed mean R0=2.9 and borrowed dispersion parameter as k=0.2 from PMC7338915, PMID: 33024095 and https://www.medrxiv.org/content/10.1101/2020.06.28.20142133v2). The main difference is that only ~60% of individuals do not produce secondary infections when the mean R0 is high in comparison to ~75% when we assumed low mean R0. These results are briefly described in the revised paper but we do not include the figure in the revisions as we do not wish to bloat the paper as it already contains a high number of figures and supplementary figures.

**Author response image 2. sa2fig2:** The distribution of individual R0, Serial Interval and Generation Time, when we assume data with higher R0 of 2. 8-2.9.

3) Distinction in assumptions for flu and covidThe populations on which the histograms for the two diseases are based are quite different. For SARS-CoV-2, the studies are from China (Shenzhen, Tianjin and Hong Kong), while those for influenza are from Switzerland. Could cultural differences be relevant? What about seasonal differences, as the time during which the early SARS-CoV-2 studies occurred was necessarily restricted?

This is a very important point as well. We have amended the Discussion to make the more conservative conclusion that the model explains the differences in these two settings for these two viruses. That said, we are extremely confident that our qualitative conclusions can be generalized for global influenza and SARS-CoV-2 for the simple reason that super-spreader events are a fundamental feature of the coronavirus in every country where it has been assessed, whereas these events are very rarely reported for influenza. We make this point in the revised Discussion.

We also repeatedly emphasize that the estimate of our 4 parameter values are bound to differ in different countries with different cultural practices and contact matrices.

Furthermore, the explanation for the difference between influenza and COVID is based primarily on differences in contact patterns. While the Discussion clarifies this to be based on the efficiency with which exposures lead to infections (and pre-symptomatic transmission), which does sound like a viral parameter, rather than a social one. These viral factors do seem more believable than having to explain why the patterns of social contact exhibited by influenza patients would differ from those of SARS-CoV-2 patients. More focus on possible mechanistic explanations is warranted.

Thank you. It is a subtle point but we feel that the contact network parameters that differ between the two viruses are dictated less by social factors and more by critical differences in viral biology. First, both viruses spread in very similar populations. Moreover, as stated in the paper, if a SARS-CoV-2 infected person and an influenza infected person (both at peak viral loads) are placed in an equivalent crowded and poorly ventilated room, the SARS-CoV-2 infected person will have the potential to have exposure contacts with a greater proportion of people with the virus. The most parsimonious explanation is aerosolization for SARS-CoV-2 being much more important than for influenza. We now conclude the paper with a conceptual figure (Figure 10) which illustrates this point.

Title: "Contact heterogeneity" rather than "contact network" is more appropriate for the title as no network is considered.

We absolutely agree. Thank you for this point.

[Editors' note: further revisions were suggested prior to acceptance, as described below.]

Revisions:A key definition in the work (as stated in the Discussion) is the exposure contact. However, apart from the parenthetical definition in the Discussion, exposure contact is not clearly defined. This is particularly important as estimates on the exposure contact seem to impact the estimates on the viral load to infectiousness functional relationship. In particular, infectiousness is defined "… as the viral load dependent probability of transmission given direct airway exposure to virus in an exposure contact." Why does the parameterization for this change so significantly pre and post lockdown when the main difference appears to be in exposure contacts?

Thank you for pointing this omission out. We have now added the following definition: an exposure contact is a susceptible person who is exposed to a SARS-CoV-2 infected person for a sufficient period of time and at a close enough distance to allow for the possibility of a successful transmission. The probability of transmission then becomes dependent on viral load of the infected person. For this reason, the number of exposure contacts may go down under conditions of quarantine or self-isolation upon developing symptoms.

The explanation given refers to "… more prolonged and intense exposure contacts…" This seems to imply an unequal reflection of what an "exposure contact" truly means. Furthermore, how does the "exposed contact rate" compare with exposure contacts.

We emphasize that exposure contacts were likely different (longer and more intimate) prior to widespread knowledge of the virus in Wuhan allowing for lower viral load transmissions. We eliminate the paragraph with “exposed contact rate” as we agree this is confusing.

Figure 10. This new figure is confusing. In the text, viral shedding kinetics are referred to as panel (A) and kinetics of infectivity as panel (B), but they are both part of panel (A) in the figure. What is different between the SARS-CoV-2 and Influenza schematics in panel (B). Panel (C) is referred to in the text but is not in the figure.

We fixed the paper such that the text matches the paper.